# Improved spectral processing for a multi-mode pulse compression Ka/Ku-band cloud radar system

Han Ding[1,2], Haoran Li[2,3], Liping Liu[2]

[1]Collaborative Innovation Center on Forecast and Evaluation of Meteorological Disasters, Nanjing University of Information Science and Technology, Nanjing, China

[2]State Key Laboratory of Severe Weather, Chinese Academy of Meteorological Sciences, Beijing, China

[3]Institute for Atmospheric and Earth System Research/Physics, Faculty of Science, University of Helsinki, Helsinki, Finland

*Correspondence to*: Haoran Li (lihr@cma.gov.cn) and Liping Liu (liulp@cma.gov.cn)

**Abstract.** Cloud radars are widely used in observing clouds and precipitation. However, the raw data products of cloud radars are usually affected by multiple factors, which may lead to misinterpretation of cloud and precipitation processes. In this study, we present a Doppler-spectra-based data processing framework to improve the data quality of a multi-mode pulse-compressed Ka/Ku radar system. Firstly, non-meteorological signal close to the ground was identified with enhanced Doppler spectral ratios between different observing modes. Then, for the Doppler spectrum affected by the range sidelobe due to the implementation of the pulse compression technique, the characteristics of the probability density distribution of the spectral power were used to identify the sidelobe artifacts. Finally, the Doppler spectra observations from different modes were merged via the shift-then-average approach. The new radar moment products were generated based on the merged Doppler spectrum data. The presented spectral processing framework was applied to radar observations of a stratiform precipitation event, and the quantitative evaluation shows good performance of clutter/sidelobe suppression and spectral merging.

## 1. Introduction

Clouds and precipitation are important for the Earth's energy budget and the hydrological cycle. Over the past few decades, a great deal of effort has been made to understand the microphysics and dynamics of clouds and precipitation. As remote sensing instruments, cloud radars operating in millimeter-wavelengths have shown their unique role in addressing the observational gaps in clouds and precipitation (Kollias et al., 2002; Stephens et al., 2002; Illingworth et al., 2007; Li and Moisseev, 2020). Compared with weather radars, shorter wavelengths of cloud radars allow for the

detection of small hydrometeors without the use of high-power transmitters and large antennas. Meanwhile, their compact size enables good portability, making them a powerful tool for observing clouds and weak precipitation (Kollias et al., 2007).

Most cloud radars work at the vertically pointing mode, and it is a common practice to use time-height plots to present the traditional radar data, such as equivalent reflectivity factor ($Z_e$), mean Doppler velocity ($V$), and spectrum width ($\sigma$). These data products are also known as the moments of radar Doppler spectrum which is the decomposition of the radar return as a function of Doppler velocities (Kollias et al., 2011a). Radar Doppler spectra observations have been used to retrieve the dynamics (Shupe et al., 2008; Li et al., 2021; e.g., Zhu et al., 2021) and microphysics (Luke and Kollias, 2013; Tridon et al., 2013; Verlinde et al., 2013; Kalesse et al., 2016; e.g., Kneifel et al., 2016) of clouds and precipitation. However, preprocessing of radar Doppler spectra observations can be challenging due to a list of issues:

1) The contamination of non-meteorological signals. The non-meteorological echoes produced by stationary targets (e.g., buildings, trees or terrain) and moving targets (e.g., insects, birds, or power lines moving in the wind) are unwanted but often detected by radar. Narrow-beam-width antenna makes the cloud radars less susceptible to non-meteorological signals in contrast to high-power long-wavelength radars (Kollias et al., 2007). To discriminate clutter echoes from clouds, some algorithms, e.g., based on the coherent characteristics of clouds (Kalapureddy et al., 2018), the Bayesian method (Hu et al., 2021), or polarimetric measurements (Martner and Moran, 2001), have been proposed. But such approaches fall short when meteorological signals are mixed with clutter. Alternatively, cloud/precipitation signals can be discriminated from clutter properly if the clutter removal is made in the radar Doppler spectrum (Luke et al., 2008; Moisseev and Chandrasekar, 2009; Williams et al., 2018; Williams et al., 2021). For example, for stationary ground clutter signals characterized by the Doppler velocity of around 0 m s$^{-1}$, an interpolation method can be performed to remove the clutter after identifying the narrow spectral peaks (Williams et al., 2018). Williams et al. (2021) have also used spectral linear depolarization ratio observations to identify asymmetric insect clutters. To the best of our knowledge, there is a lack of a non-polarimetric spectral approach to separate such non-stationary clutter signals.

2) The advance in solid-state amplifiers has led to the development of solid-state cloud radars.

Solid-state transmitters are typically smaller, more reliable and more affordable than traditional vacuum tube type transmitters, but their output power is much lower than other types of tubes. To enhance the detection capability, modulated wide pulses are transmitted and then compressed into short pulses after being received. The pulse compression techniques are widely employed to achieve high range resolutions, however, significant range sidelobe can be present around radar echoes. This

may have a negligible impact on $Z_e$, but can severely affect the estimation of higher-order radar moments (Liu and Zheng, 2019). To remove the sidelobe artifacts introduced by the pulse compression, a simple threshold approach (Moran et al., 1998; Clothiaux et al., 1999) has been applied to radar moment products. To alleviate the range sidelobe contamination, the processors of Atmospheric Radiation Measurement (ARM) Millimeter Wavelength Cloud Radars (MMCRs) have

been upgraded by reducing the number of code bits used in pulse-compressed modes (Moran et al., 2002). In China, pulse compression cloud radars are nationally deployed and sidelobe contamination is one of the major issues in radar data products. The threshold approach has been applied to the Doppler spectrum observations by Liu and Zheng (2019). However, the best power threshold always needs to be adjusted according to the received signal, and sometimes several rounds of processing

are required.

3) Multiple operating modes have been employed to address the trade-off among the sensitivity, spatial and temporal resolution, Nyquist velocity, and maximum unambiguous range. For modes with pulse compression techniques, the emission of long pulses leads to an increase in radar blind range, limiting the capability of mapping the vertical distributions of clouds. However, the blind

zones and sensitivities of different observing modes are different, leaving complicated data processing procedures in radar applications.

In this study, we present an improved data processing framework to tackle the above-mentioned issues. Section 2 describes the radars used in this study, followed by clutter and sidelobe artifact removal algorithms in Sect. 3. The merging of Doppler spectra observations at different modes is

given in Sect. 4. The new data processing framework was applied to radar observations of a stratiform precipitation event and the results are quantitative evaluated in Sect. 5. Conclusions are presented in Sect. 6.

**2. Data**

The vertically pointing Ka/Ku dual-frequency radar used in this paper has been operating at

the Longmen Observation Station (114.27°E, 23.79°N, 80.3 m above mean sea level) in southeastern

China since 2019. The operating parameters of four observation modes are shown in Table 1. Both

radars are implemented with solid-state transmitters and pulse compression techniques. The

maximum detection range is 15 km and the range resolution of 30 m. The antenna beamwidth is 0.9°

for the Ku-band radar and 0.35° for the Ka-band. Both radars operate with four modes: boundary

layer mode (mode 1), cirrus mode (mode 2), precipitation mode (mode 3), and middle-level mode

(mode 4). These four modes are characterized by different pulse compression ratios, numbers of

coherent integration as well as incoherent integration. The boundary layer mode aims to detect low-

level clouds and a narrower pulse waveform as well as a larger number of coherent integrations is

used to improve the detection ability. The cirrus mode uses the pulse compression technique to

improve the sensitivity to detect clouds with weaker radar echoes at higher altitudes. The middle-

level mode also uses pulse compression techniques but less coherent integration times. The

precipitation mode is characterized by a larger unambiguous range and velocity for rainfall

observations. These four different modes are routinely cycled in operations and each mode takes 7

s to finish the observation. The radar Doppler spectra are computed using a 256-point fast Fourier

transform (FFT). The resolutions of spectral velocity at all modes are interpolated into 0.072 m s$^{-1}$

(Ka-band radar) and 0.09 m s$^{-1}$ (Ku-band radar). The spectral noise floor is determined using the

Hilderbrand-Sekhon method (Hildebrand and Sekhon, 1974). It should be noted that due to the use

of long pulses in mode 2 and mode 4 for both radars, the heights below 2 km and 1 km are blind

zones, respectively. The blind zones of modes 1 and 3 are 30 m. In addition, the Nyquist velocity of

the Ka-band radar at mode 1 is 4.6 m s$^{-1}$, and the observed Doppler spectrum easily gets aliased

therefore the Ka-band radar observations at mode 1 were not used.

The cross-calibration between different modes is necessary before comparing observations at

different modes. We selected the stable and weak precipitation cases, and the systematic offset in

reflectivity observations was identified. For both radars, the reflectivity observations at mode 2 were

used as the reference to calibrate radar data at other modes. For both radars, the reflectivity

observations at mode 2 were used as the reference to calibrate radar data at other modes. The

reflectivity offsets are 3.8 dB (mode 2 - mode 3) and -3.6 dB (mode 2 - mode 4) at Ka-band,

respectively. For the Ku-band radar, these values are 7.5 dB (mode 2 – mode 1), -1.0 dB (mode 2 – mode 3) and -2.9 dB (mode 2 – mode 4), respectively. Note that we did not do the attenuation calibration, since it is out of the scope of this study.

**Table 1.** Operating parameters for the Ka/Ku-band cloud radar system deployed at Longmen observation station in southeastern China.

| Parameters | Mode 1 Boundary layer mode | Mode 2 Cirrus mode | Mode 3 Precipitation mode | Mode 4 Middle-level mode |
|---|---|---|---|---|
| Pulse width (μs) | 0.2 | 12 | 0.2 | 6 |
| Pulse length (m) | 60 | 3600 | 60 | 1800 |
| Range resolution (m) | 30 | 30 | 30 | 30 |
| Nyquist velocity of Ka-band ($m\ s^{-1}$) | 4.63 | 9.27 | 18.54 | 18.54 |
| Nyquist velocity of Ku-band ($m\ s^{-1}$) | 11.48 | 22.97 | 45.95 | 45.95 |
| Spectral velocity resolution of Ka-band ($m\ s^{-1}$) | 0.036 | 0.072 | 0.145 | 0.145 |
| Spectral velocity resolution of Ku-band ($m\ s^{-1}$) | 0.09 | 0.18 | 0.36 | 0.36 |
| Number of coherent integrations | 4 | 2 | 1 | 1 |
| Number of incoherent integrations | 16 | 32 | 64 | 64 |
| Number of points in FFT | 256 | 256 | 256 | 256 |

### 3. Clutter and range sidelobe mitigation

Clutter contamination is a long-standing issue in scanning and vertically pointing radar observations. Both ground clutter and insect clutter obscure the boundary layer returns, affecting the high-order moments estimated from Doppler spectra observations (Sato and Woodman, 1982). In addition, the implementation of pulse compression techniques in modes 2 and 4 usually results in significant range sidelobes around the melting layer, which does not significantly affect $Z_e$ and $V$ estimates, but can severely degrade the estimation of spectrum width. In this section, Ku-band radar observations are used to demonstrate the spectral processing procedure for mitigating the clutter contamination and range sidelobe.

### 3.1 Clutter mitigation

The stationary ground clutter is usually manifested as a narrow-symmetric peak around 0 m s⁻

[1] (Williams et al., 2018). A commonly used approach for mitigating ground clutter signals is the interpolation of adjacent spectral powers after removing the spectral peak around 0 m s[-1]. Williams et al. (2018) claimed that this method is also suitable for the identification and removal of insect clutters since the insect targets also produce narrow peaks in Doppler spectra observations. We have tried to apply this approach to our radar data, but the performance is not as good as that of the Ka-

band zenith pointing radar (KAZR) deployed at Oliktok Point, Alaska. Figure 1a shows an example of the Ku-band Doppler spectrum with clutter signals present at around 0 m s[-1]. The clutter signals do not always present a sharp narrow peak as shown in Fig. 3 in Williams et al. (2018), and this approach does not apply to our observations. We have also found that such clutter signals appear more frequently and significantly in Ku-band radar observations than in the Ka-band.

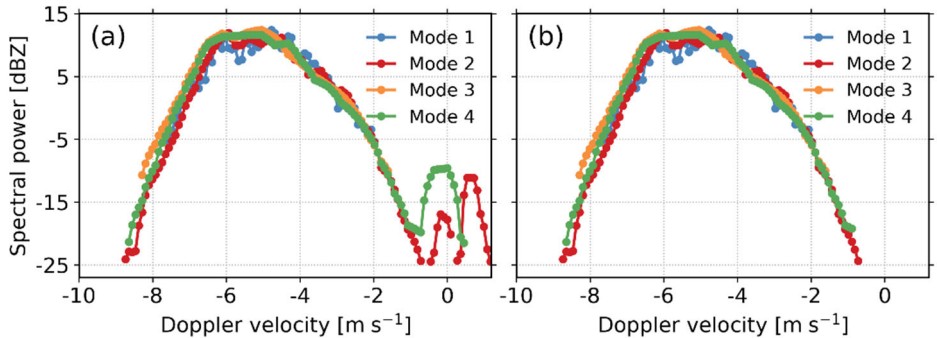


**Figure 1.** (a) Noise-removed Ku-band Doppler power spectrum on 6 June 2020 at 22:52:08 LST at the range of 2.34 km. (b) Same as (a) but decluttered with our clutter mitigation algorithm. The unit of Doppler power spectral density data is $mm^6 \, m^{-3} \, (m \, s^{-1})^{-1}$, we simply use the unit "dBZ" here and after to denote spectral power in the dB scale.


Figure 2 shows the time series of Doppler velocity spectra on 6 June 2020 from 22:40 to 23:01 LST at 2.34 km range (the same range bin as Fig. 1). The clutter signals are in the vicinity of 0 m s[-1] and are not continuous with time. Compared with meteorological signals, it appears that clutter echoes randomly occur with some dependence on the observing mode. The cause of such clutter

signals is unclear yet and we hesitate to attribute them to insects (Williams et al., 2018) since the spectral powers at different modes deviate from each other significantly.

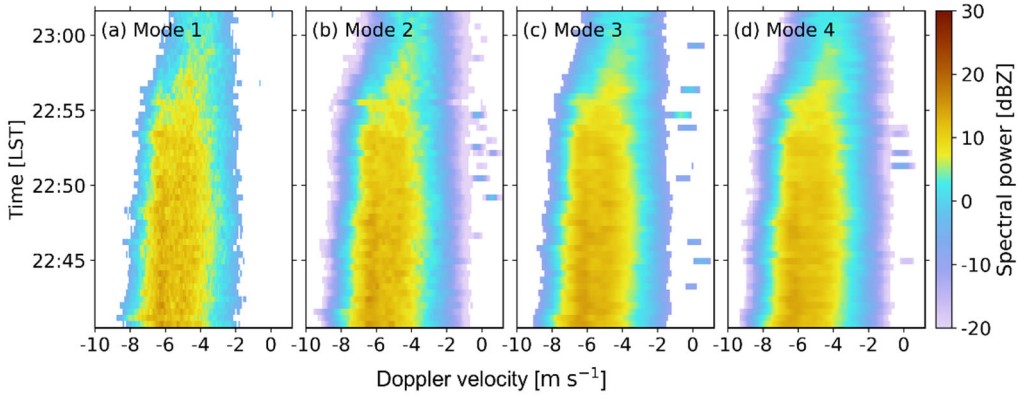

**Figure 2.** Time series of noise-removed Ku-band Doppler velocity spectra on 6 June 2020 from 22:40 to 23:01 LST at 2.34 km range.

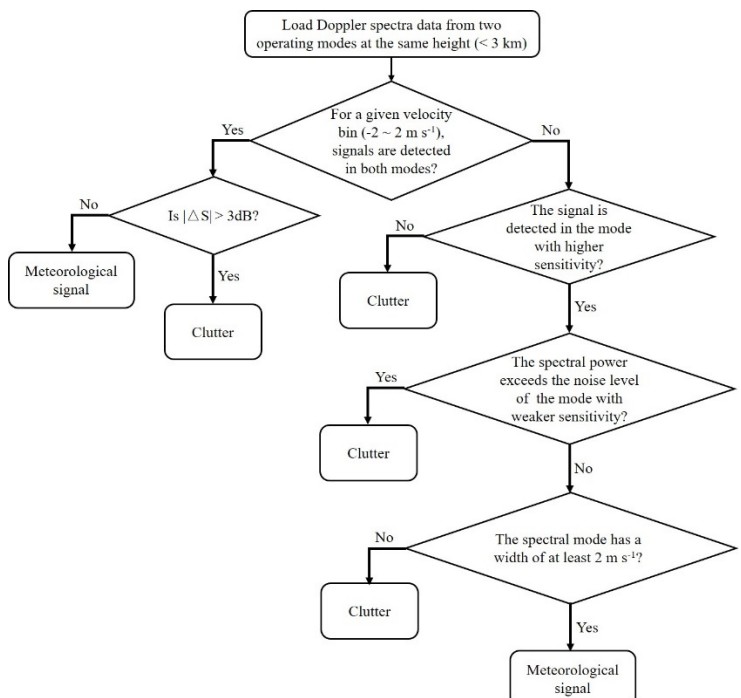

**Figure 3.** Flow chart of the proposed clutter identification and mitigation method.

As shown in Fig. 3, we have developed an algorithm to identify and remove clutter signals. The algorithm is mainly based on the non-coherent nature of clutters which produces significant spectral power ratio ($\Delta S$) between observations from different modes. The selection of the threshold is a comprise between false-alarm and miss hit. We want to preserve the meteorological signals at our best, therefore we checked the magnitudes of $|\Delta S|$ for meteorological signals. Appendix A presents the statistical plot of $|\Delta S|$ for meteorological signals (height of $2 \sim 3$ km and Doppler velocity of $2 \sim 5$ m s$^{-1}$). It appears that the probability of $|\Delta S|$ tends to be flat after 3 dB, and the use

of 3 dB can ensure that 95.6 % of precipitation signals are well preserved (Fig. A1). Therefore, 3 dB is used in this study. If a larger threshold is employed, we expect more clutter signals will be mislabeled as precipitation. As shown in Fig. 1b, clutter signals have been successfully removed while the meteorological signals are marginally affected.

175       It should be noted that this method relies on observations recorded at different observing modes. However, the sensitivities of different modes are not identical. Therefore, if the clutter is presented in the most sensitive mode (e.g., mode 2) only, it cannot be filtered out with the $|\Delta S|$ method. In this case, the width of valid meteorological spectral mode is assumed to be longer than 2 m s$^{-1}$, otherwise it is attributed to clutter. We are aware that Shupe et al (2004) have used a width of 0.448 m s$^{-1}$ to

180       identify supercooled liquid water. We have tried this value, but the width of clutter present in this dual-wavelength radar system easily exceeds 1 m s$^{-1}$ (Fig. 2). Actually, the selection of the spectrum width is similar with the use of a signal-to-noise ratio (SNR) value in noise-removal. Higher SNR means a stricter noise-removal but higher chance of losing valid signals. We have tested the width of 1, 1.5, 2, and 3 m s$^{-1}$ (visual inspection, not shown), and found that 2 m s$^{-1}$ can effectively remove

185       clutter signals for both radars though very light precipitation (detected by the most sensitive mode only) can be removed as well. Admitting this potential issue, it suffices the application in rainfall. In addition, for clouds with highly variable reflectivity, the presented algorithm may mislabel them as clutter according to our assumption that meteorological signals are coherent in a round of observation (28 s).

190       Figure 4 compares the Doppler spectrum observations before and after applying the declutter algorithm. As shown in Fig. 4a$_1$ and c$_1$, the clutter signals appear below 2 km at modes 1 and 3. For modes 2 and 4, the impact of clutter can be up to 3 km (Fig. 4b$_1$ and d$_1$). After imposing the declutter algorithm, no significant clutter signals can be detected (Fig. 4a$_2$, b$_2$, c$_2$, and d$_2$).

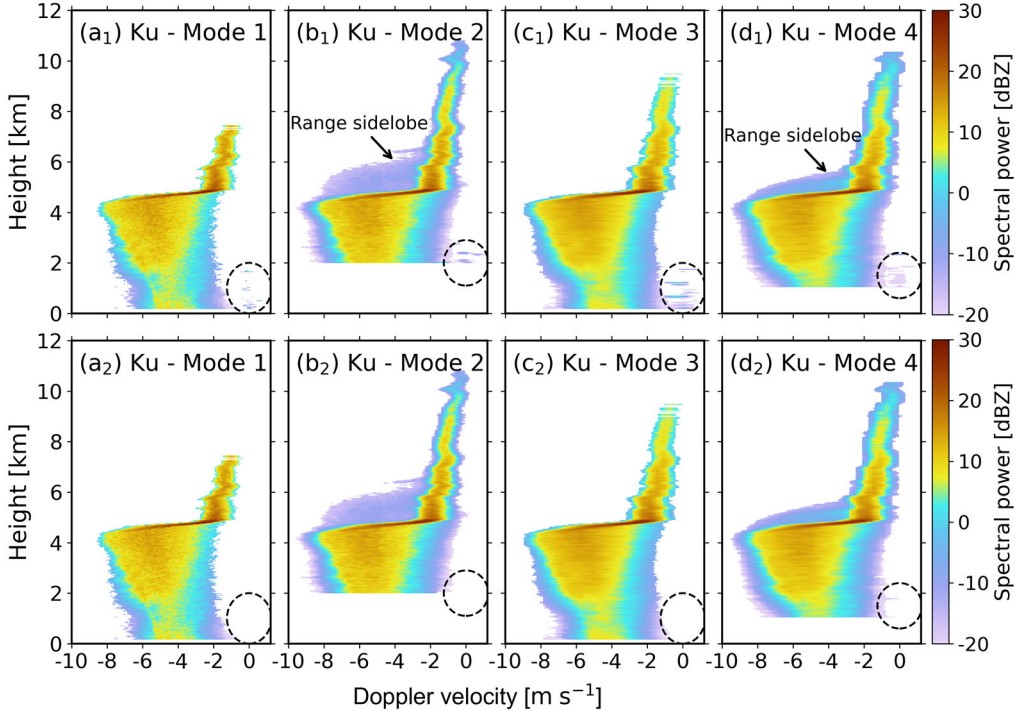

Figure 4. Top: noise-removed Ku-band Doppler power spectra on 6 June 2020 at 22:52:08 LST recorded at ($a_1$) mode 1, ($b_1$) mode 2, ($c_1$) mode 3, ($d_1$) mode 4. Bottom: decluttered observations. The dashed circles mark the clutter signals. Note that the heights below 2 km and 1 km are blind zones for modes 2 and 4, respectively.

## 3.2 Range sidelobe artifacts

The utilization of pulse compression usually leads to significant range sidelobe artifacts (Fig. 4$b_1$ and $d_1$) around melting layer, which can severely affect the estimates of high-order radar moments. Moran has proposed an approach that distinguishes the range sidelobe artifacts from reflectivity data using non-range-corrected return power through the power transfer function (Moran et al., 1998; Clothiaux et al., 1999). By reducing the number of code bits used in pulse compression modes, the ARM MMCRs' upgraded processor is capable of suppressing the range sidelobe effects (Moran et al., 2002). However, mitigating range sidelobe artifacts is still challenging for multi-mode pulsed-compression cloud radars in China. To improve both the radar detection performance and range resolution, Linear Frequency Modulation was used to widen the signal bandwidth when transmitting pulses in modes 2 and 4 at both Ka- and Ku-band. But, the matched pulse compression filter output exhibits sidelobe behavior, making the power of range sidelobe appear in the wrong range gates. Liu and Zheng (2019) have applied the method proposed by Moran et al. (1998) to

radar Doppler spectrum data to remove the range sidelobe artifacts. However, the performance of this approach depends on a given threshold, which needs to be adjusted for different scenarios.

As shown in Fig. $4b_1$ and $d_1$, the range sidelobe associated with the strong radar echoes of the melting particles is located above the melting layer. Compared with radar Doppler spectrum observations without the sidelobe contamination (see for example Li and Moisseev, 2020), Doppler spectra above the melting layer at large velocity bins were contaminated by the range sidelobe of the echo below. The artifacts in mode 2 accumulate to higher altitudes but are weaker in spectral power (Fig. $4b_1$), while mode 4 accumulates to lower altitudes and with a larger magnitude of power (Fig. $4d_1$), which is caused by different pulse compression ratio and peak sidelobe ratio (the ratio of the main lobe peak power to the highest sidelobe peak power) of the two modes.

An interesting feature of the range sidelobe caused by pulse compression is that its spectral power is much flatter than cloud and precipitation signals. Figure 5a shows the probability density functions (PDFs) of received spectral power at 2.4 km, 5.01 km, and 6.6 km, which respectively represent the liquid precipitation, Doppler spectrum contaminated by range sidelobe, and solid precipitation. For the sidelobe-contaminated Doppler spectrum, it can be seen that the range bins contaminated by range sidelobe have different spectral power distributions, the peak of the PDFs appears close to the noise level and is mostly below 15 dB above the noise level. A closer look into the radar Doppler spectra at 5.01 km (Fig. 6a) shows that the strong PDF peak in Fig. 5b is explained by the relatively flat range sidelobe signals. Here, we introduce a parameter spectral power threshold ($S_{thresh}$) to distinguish the range sidelobe from meteorological signals. Figure 7 shows the flowchart for the identification and removal of the range sidelobe artifacts. The procedures are briefly summarized as follows:

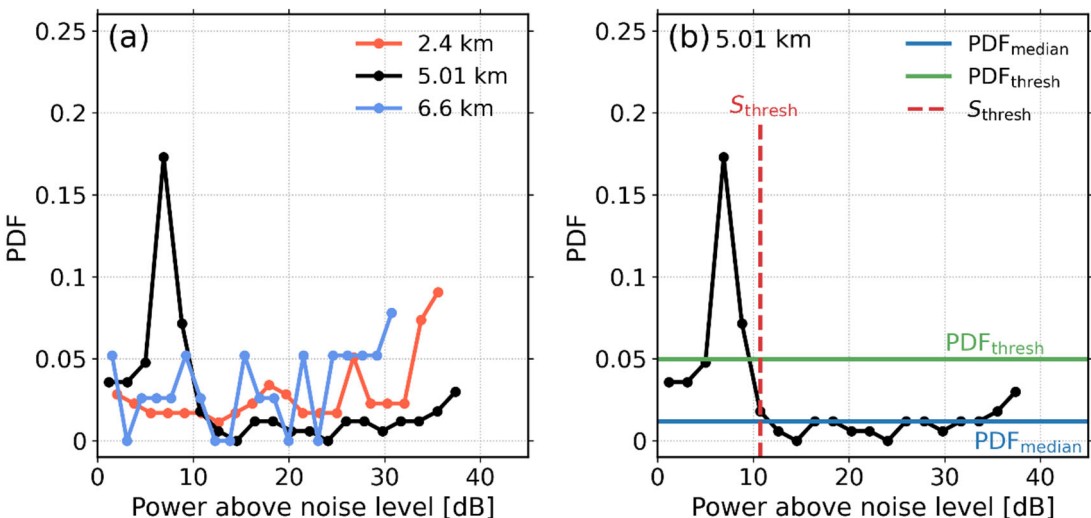

**Figure 5.** (a) PDFs of Doppler spectra from 2 km to 7 km at mode 2; (b) PDF of Doppler spectrum recorded at 5.01 km.

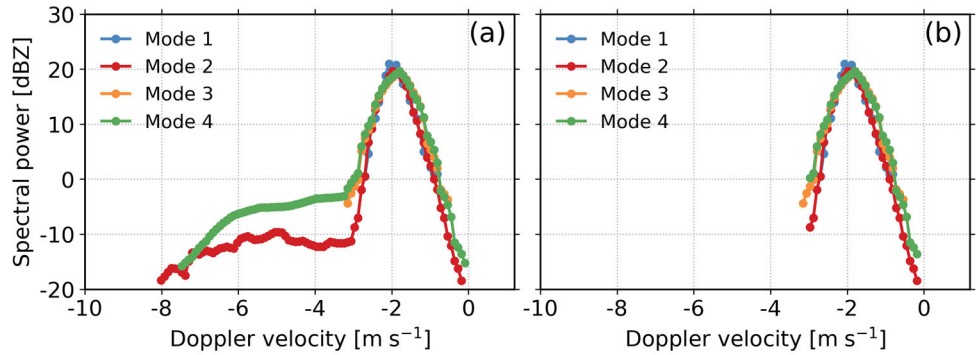

**Figure 6.** Ku-band Doppler power spectra recorded at different modes at 5.01 km on 6 June 2020 at 22:52:08 LST. (a) Noise-removed Doppler spectrum; (b) the same as (a) but after the removal of range sidelobe.

1) Sort the spectral power values above noise level in an ascending order to get a PDF curve of each Doppler spectrum;

2) Calculate the median and standard deviation (SD) of the PDFs, set $PDF_{thresh} = PDF_{median} + PDF_{SD}$; Note that the determination of this relation is given in Appendix B.

3) Below half of the peak power above the noise level of the Doppler spectrum, find the power bins' probability density just exceeds the $PDF_{thresh}$, and the corresponding spectral power is set as $S_{thresh}$; (The range of $PDF_{thresh}$ is limited to half of the peak power above the noise level to avoid finding the $PDF_{peak}$ corresponding to large spectral power, which makes the determined $S_{thresh}$ corresponds well to the power of sidelobe in this way.)

4) If the spectrum power with the Doppler velocity larger than the mean Doppler velocity is

below the $S_{thresh}$, then it is flagged as sidelobe.

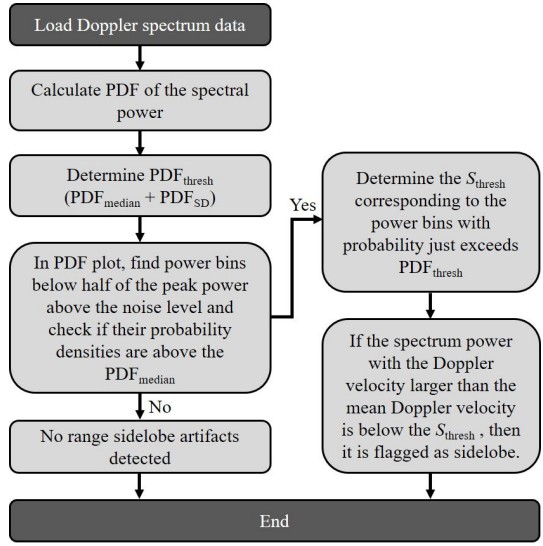

**Figure 7.** Flowchart of range sidelobe artifacts processing.

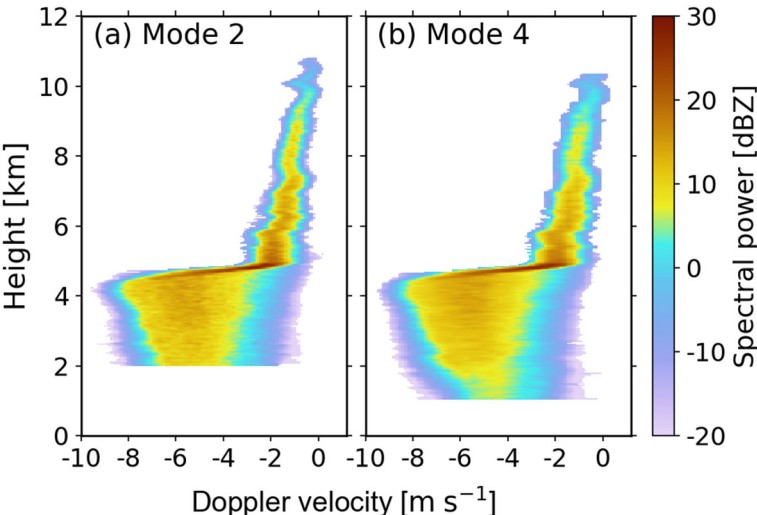

**Figure 8.** Doppler power spectra after removing range sidelobe at modes 2 and 4 on 6 June 2020 at 22:52:08 LST. The Doppler spectra observations before the sidelobe removal are shown in Fig. 4b$_1$ and d$_1$.

As shown in Fig. 6b, the range sidelobe artifacts in modes 2 and 4 have been well removed. We have applied this algorithm to the vertical profiles of Doppler spectra observations at modes 2 and 4 (Fig. 4b$_1$ and d$_1$). As shown in Fig. 8, the sidelobe artifacts have been well removed at modes 2 and 4. Furthermore, we have compared this algorithm with the threshold method (Liu and Zheng, 2019), all the results and analysis are included in Appendix C.

**4. Mode merging**

For multi-mode cloud radars, it is cumbersome to interpretate the radar observations recorded

at four modes in operational applications. Moreover, the air motion variability and the velocity bin-to-bin spectrum power fluctuations can lead to noisy estimates of high-order spectral moments. Therefore, we have merged radar observations from different observing modes. Data from Ku-band was still taken as an example to illustrate the data merging process.

**4.1 Merging of Doppler spectra recorded at different modes**

Before the merging procedure, it is necessary to check the consistency of radar data at four modes. Particularly, coherent integrations were made to modes 1 and 2 (Tab.1) to improve the signal-to-noise ratio. But this step may result in the decrease of spectral power with large Doppler velocities (Liu et al., 2017; Liu and Zheng, 2019). This effect leads to the underestimation of $V$, which is critical in the merging process, and $Z_e$. Here, we evaluate this impact by comparing $Z_e$ and $V$ estimates at different modes. We define the differences of $Z_e$ and $V$ between different modes as $\Delta Z$ and $\Delta V$, respectively, and radar observations at mode 3 (no pulse compression and only one round coherent integration were performed) were used as a reference. To compare the impact of coherent integration under various precipitation intensities, radar observations were grouped into $Z_e > 20$ dBZ and $Z_e < 10$ dBZ. Note that the Ku-band wet radome attenuation has been corrected with a collocated C-band radar (Cui et al., 2020).

In light precipitation ($Z_e < 10$ dBZ, Fig. 9$a_1$, $b_1$, and $c_1$), radar observations at these four modes agree with each other rather well. For precipitation cases with $Z_e > 20$ dBZ, good agreement between modes 3 and 4 can also be found (Fig. 9$c_2$), which is expected since the coherent integration number is one at both modes. The agreement between modes 2 and 3 seems also good (Fig. 9$b_2$), despite two rounds of coherent integration being made to mode 2. In Fig. 9$a_2$, significant biases of $\Delta Z$ and $\Delta V$ can be identified, and $\Delta V$ increases with $\Delta Z$. This is attributed to the underestimation of spectrum powers at high Doppler velocities during the longtime coherent integration (4 rounds). Given the results above, Ku-band Doppler spectra observations at mode 1 were discarded.

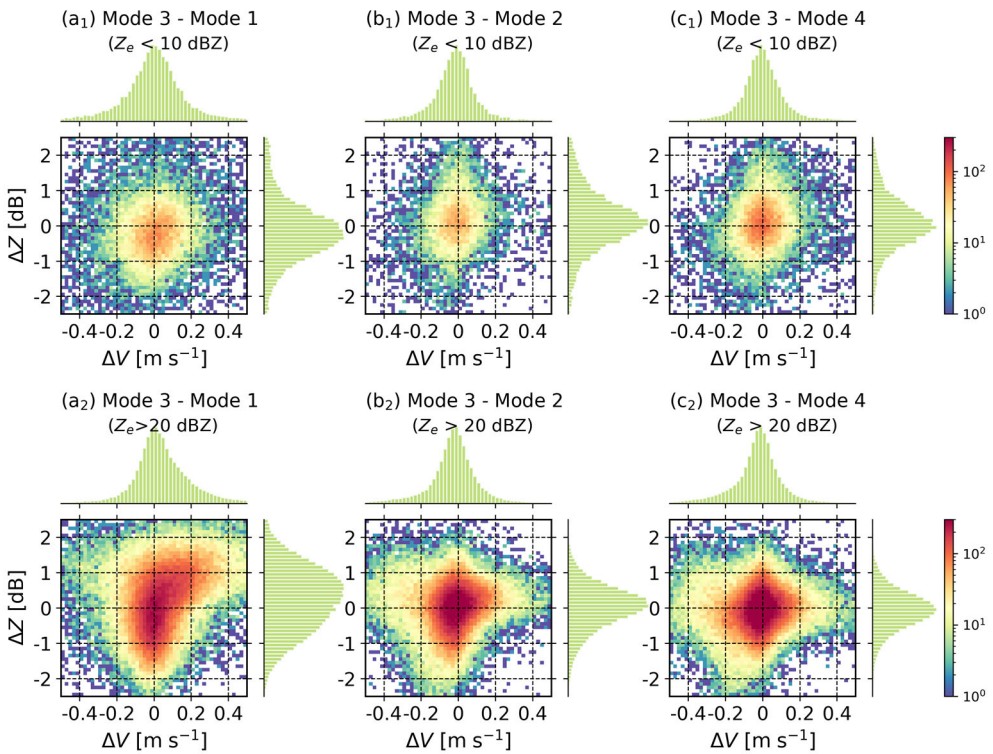

**Figure 9.** Statistics of $\Delta Z$ and $\Delta V$ for the Ku-band radar. Top: precipitation cases with $Z_e < 10$ dBZ; Bottom: precipitation cases with $Z_e > 20$ dBZ.

The same method was applied to the Ka-band radar (Fig. 10). Note that the data from mode 1 was not used due to the small Nyquist velocity (4.63 m s$^{-1}$ as shown in Tab. 1). Interestingly, 2 times of coherent integration marginally affects $\Delta Z$ and $\Delta V$ for the Ku-band radar data (Fig. 9b$_2$), but this impact is rather significant at Ka-band (Fig. 10a$_2$). Therefore, Ka-band radar observations from both modes 1 and 2 were not used.

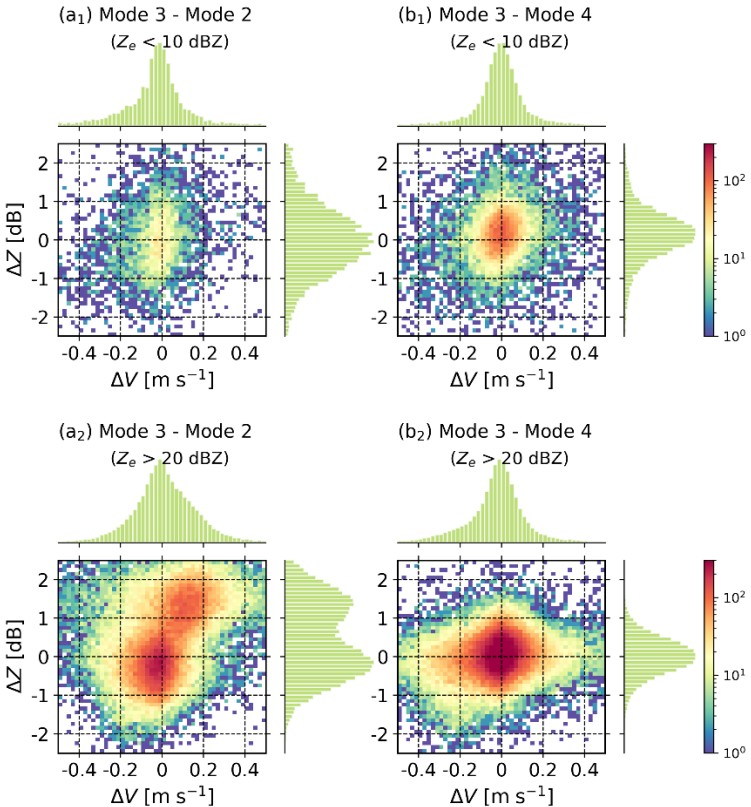

**Figure 10.** Same as Fig. 9 but for Ka-band radar. Note that the data at mode 1 was excluded due to the limited Nyquist velocity.

### 4.2 Shift-then-average spectra

To maximize the detection advantages of each mode, and to obtain high-quality and easy-to-use radar datasets, Doppler spectra observations from various modes were merged as follows (Giangrande et al., 2001; Luke and Kollias, 2013; Williams et al., 2018):

1) Velocity shift: set the mean of the mean Doppler velocity at each mode as the reference velocity, and then shift the Doppler spectrum at each mode to match the mean Doppler velocities at all modes;

2) Spectral power average: average the spectral powers observed at all modes in each observation round.

For the Ku-band radar, observations at modes 2, 3, and 4 were merged (Fig. 11a), while modes 3 and 4 were used for Ka-band radar (Fig. 11b). The merged Doppler spectrum is significantly less uncertain thanks to the averaging process. It should be noted that the drawback of the mode merging is that the time resolution changes from 7 s to 28 s.

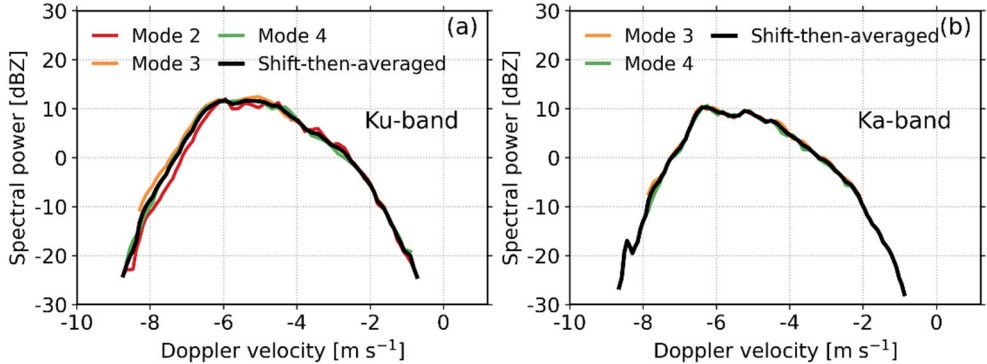

**Figure 11.** (a) Ku-band Doppler velocity spectra from modes 2, 3, and 4 recorded on 6 June 2020 at 22:52:08 LST at 2.34 km. The merged Doppler spectrum was derived from the Doppler spectra recorded at modes 2, 3, and 4 after shifting and averaging. Note that Ku-band radar observations at mode 1 were not used due to the power loss during coherent integration. (b) The same as in (a) but for Ka-band and observations at modes 3 and 4 were used.

High-order moments of the Doppler spectrum are representative of the key microphysical processes in clouds and precipitation (Luke and Kollias, 2013; Maahn and Löhnert, 2017; Li et al., 2021). The second, third, and fourth moments of the radar Doppler spectrum are spectrum width, skewness, and kurtosis respectively. Figure 12 compares these high-order moments estimated from the Ku-band radar Doppler spectra at modes 2, 3, 4, and the merged data. The sidelobe impacts on spectrum width, skewness, and kurtosis are significant between 5 and 7 km at modes 2 and 4 (Fig. 12a, b, c). In rain, the estimates of high-order moments at modes 2, 3, and 4 agree rather well with each other. In snow, the spectrum width at mode 2 is systematically smaller than those at other modes (Fig. 12a). This may be explained by the finer spectral velocity resolution at mode 2 (Tab. 1). In addition, as the radar echo approaches the noise level, underestimation of kurtosis becomes more significant (mode 3 in Fig. 12c).

The results for the Ka-band radar are shown in Fig. 13. The agreement among different modes is better than that at Ku-band thanks to higher spectral velocity resolution and less uncertainties for the Ka-band radar, while the bias of kurtosis in the snow at mode 3 (Fig. 13c) is more contrasting. These findings indicate that the uncertainties of estimated radar moments as introduced by different observing modes should be taken into account in snow retrievals (Maahn and Löhnert, 2017).

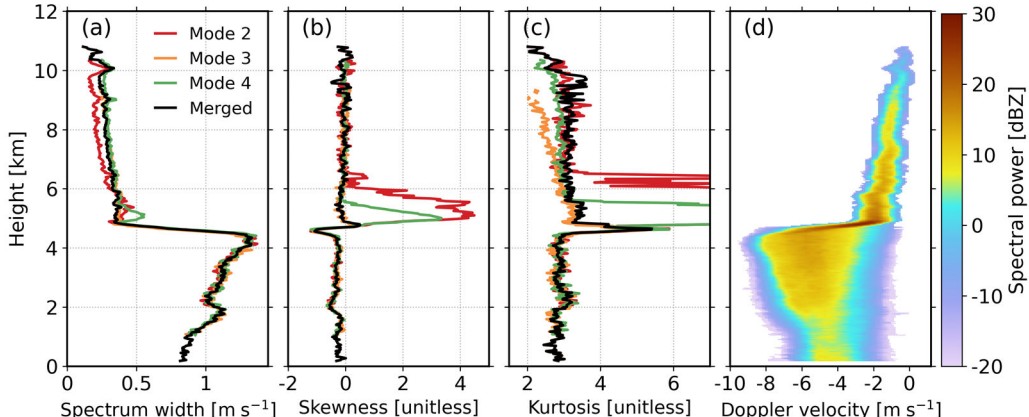

**Figure 12.** (a) Spectrum width, (b) skewness and (c) kurtosis estimated from Ku-band radar Doppler spectra recorded at modes 2, 3, 4 and the merged data; (d) the profile of merged Doppler velocity spectra. Note that Ku-band radar observations at mode 1 were not used due to the power loss during coherent integration.

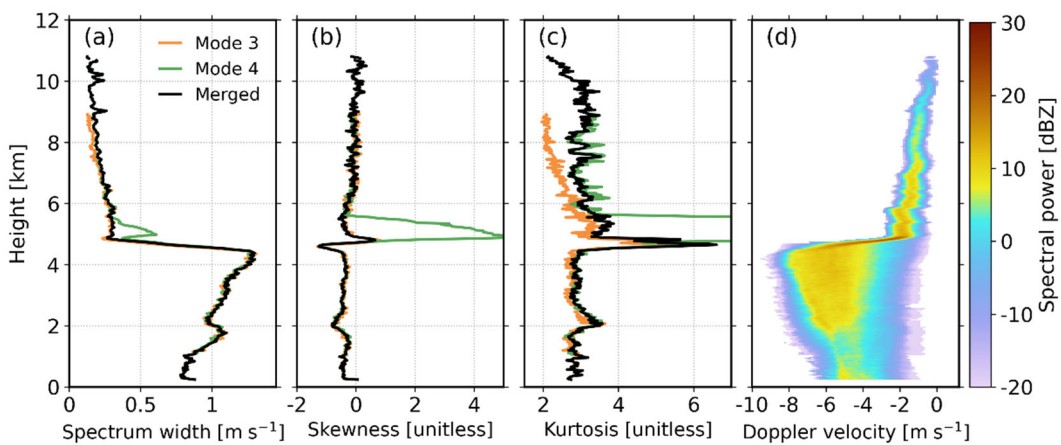

**Figure 13.** Same as Fig. 12 but for Ka-band radar. Note that Ka-band radar observations at mode 1 were not used due to the limited Nyquist velocity, while mode 2 data was discarded due to the power loss during coherent integration.

## 5. Evaluation: a case study

The presented methods were used to construct a new spectra-based radar data processing framework as shown in Fig. 14. In this section, we take a rainfall event to illustrate the algorithms presented in this study. On 6 June 2020, a stratiform rainfall system moved over the Longmen station. The melting layer is about 5 km, and the bright band signatures can be well identified from Ku- and Ka-band radar reflectivity observations as shown in Fig. 15 and 16.

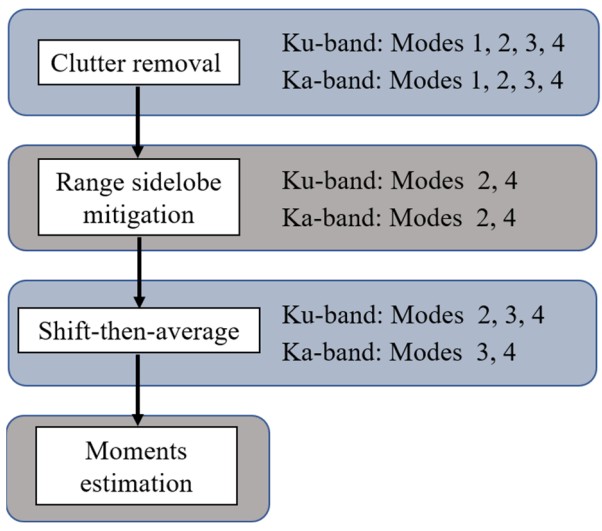

**Figure 14.** Procedures for generating estimates of spectral moments

### 5.1 Case study

To evaluate the performance of the presented framework, the merged radar products were compared with raw data products at modes 3 and 4 after processing the Ku- and Ka-band data with the proposed algorithm. The time-height cross-section plots of Ku- and Ka-band radar observations are shown in Fig. 15 and 16, respectively. The cloud top height is about 14 km (Fig. 15a$_1$), while it is much lower at mode 3 (9 ~ 10 km) which is attributed to the lower sensitivity at this mode. At

Ka-band, due to the increased attenuation from rain, melting layer, and the wet radome (Li and Moisseev, 2019), the observed cloud top descends to around 7 km during the most intensive precipitation period (around 22:00 LST, Fig. 16a$_2$). The bias in spectrum width, skewness and kurtosis introduced by sidelobe effect is rather significant at mode 4 for both radars (Fig. 15c$_3$, d$_3$, e$_3$ and Fig. 16c$_3$, d$_3$, e$_3$), while it was well mitigated in the merged products (Fig. 15c$_1$, d$_1$, e$_1$ and

Fig. 16c$_1$, d$_1$, e$_1$).

In addition, we have calculated statistics of the power leakage to range sidelobe, and the results for Ku-/Ka-band radars are given in Appendix D (Fig. D1). The results show that the sidelobe signals are usually below -20 dB. Since the reflectivity enhancement in the melting layer usually does not exceed 10 dB (Li et al., 2020), the sidelobe contamination in rain is not significant. However, the

fall velocity of snow is much slower than rain drops. Namely, no meteorological signals are present in the range of 3 ~ 10 m s$^{-1}$ and the sidelobe signal becomes evident.

Skewness and kurtosis are indicative of the degree of asymmetry and peakness of the spectrum, respectively. Skewness has been used as an early qualitative predictor of drizzle onset in clouds and

locating supercooled liquid water since it is very sensitive to drizzle generation (Luke et al., 2010;

Kollias et al., 2011a; Kollias et al., 2011b). The higher-order radar moments have been less frequently used for studying the melting layer. It appears that skewness presents a "decrease-increase-decrease" feature, while kurtosis is characterized by a distinct enhancement. These observations of skewness and kurtosis in the melting layer are interesting, and how these changes are linked to the change of cloud/precipitation microphysics warrants future studies.

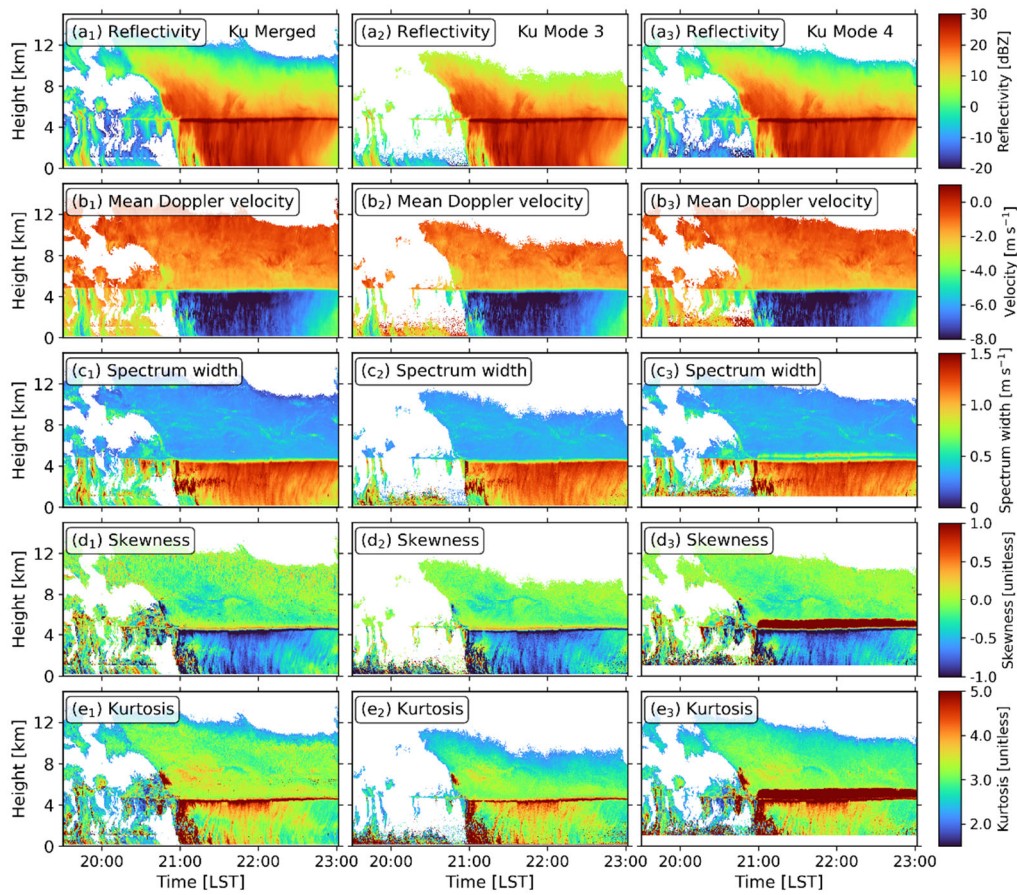


**Figure 15.** Time-height cross-section plots of Ku-band Doppler spectra moments from 19:30:19 to 23:01:26. The left column is estimated from the merged Doppler spectra, and the middle and the right columns are from the data recorded at modes 3 and 4, respectively. From top to bottom: (a) reflectivity; (b) mean Doppler velocity; (c) spectrum width; (d) skewness; (e) kurtosis.

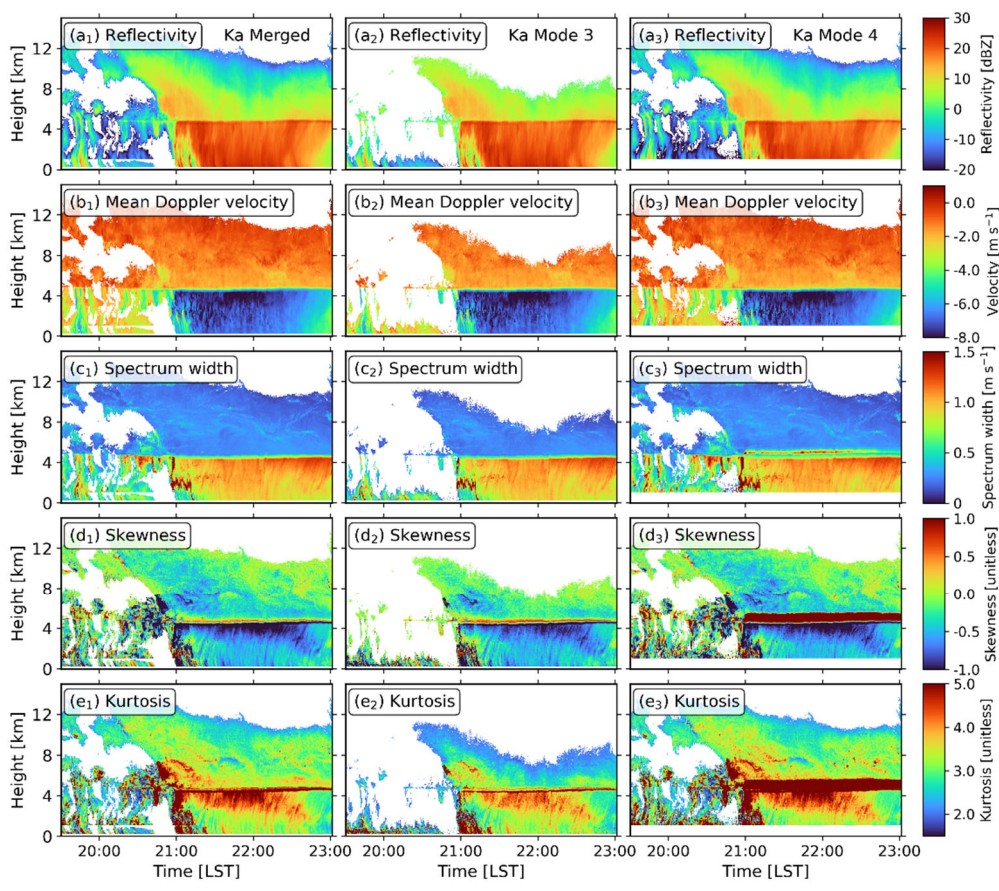


**Figure 16.** same as Fig. 15 but at Ka-band.

## 5.2 Comparison with a C-band radar

Observations from a collocated C-band frequency modulated continuous wave radar (FMCW)

radar (Cui et al., 2020) were used for a sanity check for the processed Ka/Ku-band radar data

products. The C-band radar's data products include reflectivity, Doppler velocity and spectrum

width. The spectrum width observed by the C-band radar was compared with those from mode 4 of

Ku- and Ka-band radars. As shown in Fig. 17$a_1$, $b_1$, the surge of Ku- and Ka-band spectrum width

at around 0.4 m s$^{-1}$ is attributed to the sidelobe effect, while the artifacts were well mitigated after

applying the presented algorithm (Fig. 17$a_2$, $b_2$,). It is interesting to note that the observed spectrum

width at Ku/Ka-band does not necessarily follow the 1:1 line, since the Rayleigh scattering may not

be satisfied at Ku/Ka-band for heavy precipitating cases.

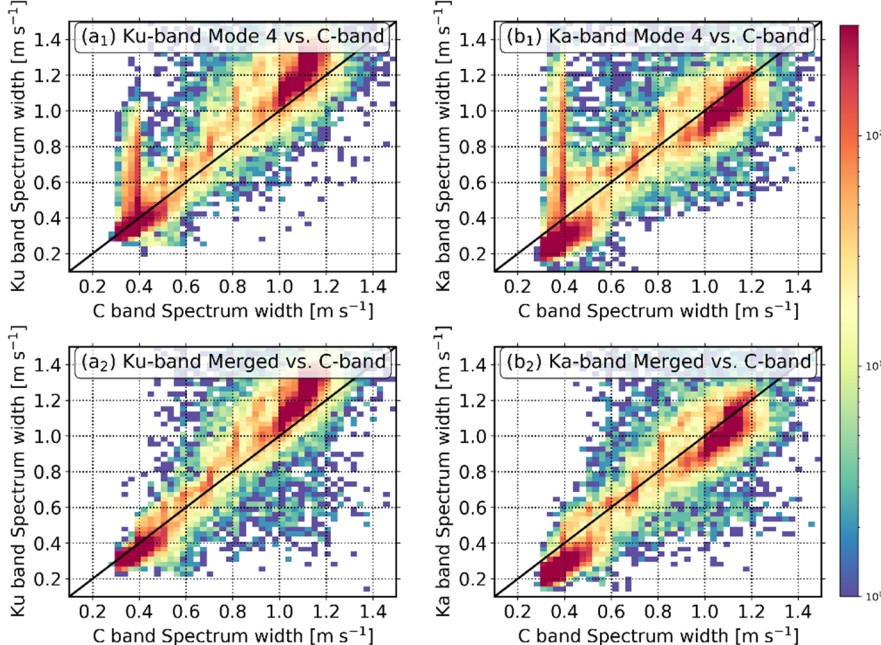

**Figure 17.** Spectrum width comparison between C-band and (a) Ku-band and (b) Ka-band. The spectrum width observations at mode 4 (upper panels) and merged data (lower panels) were used for comparison. Radar observations from 1 km to 9 km during this event were employed.

### 5.3 Quantitative evaluation

This precipitation event is also used for quantitative evaluations of the presented methods. Spectral moments (reflectivity, Doppler velocity, spectrum width, skewness and kurtosis) before and after the spectral processing are quantitatively compared to show the effectiveness and necessity of the presented methods. The results of clutter and sidelobe mitigation are presented in separate subsections.

### 5.3.1 Clutter removal

To show how the clutter mitigation procedure improves the radar data quality, we have compared the standard deviation between the data products before/after the clutter removal and the "reference data". At Ku-band, the "reference data" is defined as

$$X_{Ku,ref} = median(X_{Ku,M2}^{decluttered}, X_{Ku,M3}^{decluttered}, X_{Ku,M4}^{decluttered}) \tag{1}$$

where $X_{Ku,Mi}^{decluttered}$ denotes the spectral moment derived from decluttered Doppler spectra at mode $i$. Similarly, the "reference data" at Ka-band:

$$X_{Ka,ref} = average(X_{Ka,M3}^{decluttered}, X_{Ka,M4}^{decluttered}) \tag{2}$$

We introduce the standard deviation to assess the difference between radar products at a given mode and the "reference data":

$$SD = \sqrt{\frac{\sum_{n=1}^{n=m}(X_{Mi,n} - X_{ref})}{m}} \qquad (3)$$

where $m$ denotes the number of rainfall cases between 0 and 3 km. The results for the Ku- and Ka-band radar during this event are given in Tab. 2 and 3, respectively.

As we can see from Tab. 2, clutter signals affect the estimation of spectral moments, the SD for the reflectivity at Ku-band is reduced by a value between 0.36 and 0.8 dB after imposing the clutter removal algorithm. Significant improvement can also be identified at mean Doppler velocity and spectrum width observations. Compared with the Ku-band radar, clutter signals are weaker at Ka-band (Tab. 3). The data quality improvement of spectral moments at the Ka-band is not as pronounced as that at Ku-band, which is expected since the Ka-band radar's beam width (0.35°) is smaller than that of Ku-band (0.9°). The presented results indicate that clutter removal is essential for producing high-quality Ku data products.

**Table 2.** Standard deviation (SD) for Ku-band spectral moments before and after the declutter approach compared with the "reference data". Radar observations from 0 to 3 km are used for comparison.

| Moments | Mode 2 | | Mode 3 | | Mode 4 | |
|---|---|---|---|---|---|---|
| | SD before | SD after | SD before | SD after | SD before | SD after |
| Reflectivity (dBZ) | 0.98 | 0.62 | 1.17 | 0.37 | 1.21 | 0.56 |
| Doppler velocity (m s$^{-1}$) | 0.41 | 0.32 | 0.37 | 0.16 | 0.42 | 0.27 |
| Spectrum width (m s$^{-1}$) | 0.15 | 0.08 | 0.19 | 0.06 | 0.19 | 0.07 |
| Skewness (-) | 0.39 | 0.23 | 0.52 | 0.18 | 0.45 | 0.19 |
| Kurtosis (-) | 1.47 | 1.02 | 2.30 | 0.66 | 1.50 | 0.60 |

**Table 3.** The same as in Tab. 2 but for Ka-band.

| Moments | Mode 3 | | Mode 4 | |
|---|---|---|---|---|
| | SD before | SD after | SD before | SD after |
| Reflectivity (dBZ) | 0.51 | 0.35 | 0.44 | 0.38 |
| Doppler velocity (m s$^{-1}$) | 0.21 | 0.19 | 0.21 | 0.20 |
| Spectrum width (m s$^{-1}$) | 0.11 | 0.05 | 0.09 | 0.05 |
| Skewness (-) | 0.40 | 0.17 | 0.33 | 0.18 |
| Kurtosis (-) | 1.96 | 0.70 | 1.25 | 0.74 |

### 5.3.2 Sidelobe mitigation

The effect of sidelobe mitigation was also quantitatively evaluated. Since no pulse compression

was employed at mode 3 for the Ka/Ku-band radars, we use radar data products at mode 3 as "reference data". Radar observations from 4.5 to 6 km are used for the assessment, and the results for the Ku-band radar are given in Appendix B (see Tab. B1 for details). Since the signals associated with sidelobe are relatively weak (Fig. D1 in Appendix D), no significant changes in reflectivity and mean Doppler velocity before and after sidelobe suppression can be identified in both modes 2 and

4. As the order of spectral moments increases, the effect of range sidelobe becomes significant. The SD values of spectrum width are reduced by an order of magnitude after the sidelobe mitigation at both modes 2 and 4. Moreover, the improvement of skewness and kurtosis after the sidelobe mitigation is more obvious. The Ku-band SD of skewness at mode 2 (mode 4) decreased from 2.88 (0.4) to 1.61 (0.26), and that of kurtosis decreased from 31.38 (5.4) to 11.64 (1.32). Similar

improvement in skewness and kurtosis can also be found at Ka-band (Tab. B2 in Appendix B).

**6. Summary**

In this study, a framework for processing the Doppler spectra observations of a multi-mode pulse compression Ka/Ku cloud radar system is presented. We first proposed an approach to identify and remove the clutter signals in the Doppler spectrum based on spectral power ratios between

different operating modes. Then, we developed a new algorithm to remove the range sidelobe around the melting layer at the modes implementing the pulse compression technique. We further show that coherent integration has a decent impact on reflectivity and Doppler velocity observations and should be used with caution when the spectral merging is made. The radar observations from different modes were then merged using the shift-then-average method. The presented spectral

processing framework was applied to radar observations of a stratiform precipitation event, and the quantitative evaluations of the processed data suggest that clutter/sidelobe suppression and spectral merging results demonstrated good performance.

The presented methods mainly deal with the challenges in observing stratiform rainfall events in Southern China, given the weaker signal attenuation at both bands compared with that in

convective precipitation. We are aware that cloud radars have proven to be an effective tool for snowfall observations (e.g., Kollias et al., 2007; Li et al., 2021), the applicability of the presented framework in snowfall is expected but has not been proven yet. The multi-year radar observations

recorded at the Longmen station will be processed with the present framework for elucidating the dynamics and microphysics of clouds and precipitation in southern China.


## Appendix A: Statistics for spectral power ratios of meteorological signals between different modes at Ku-band

The meteorological signals with a height of 2 ~ 3 km and Doppler velocity of 2 ~ 5 m s$^{-1}$ were statistically analyzed to determine the appropriate $|\Delta S|$.

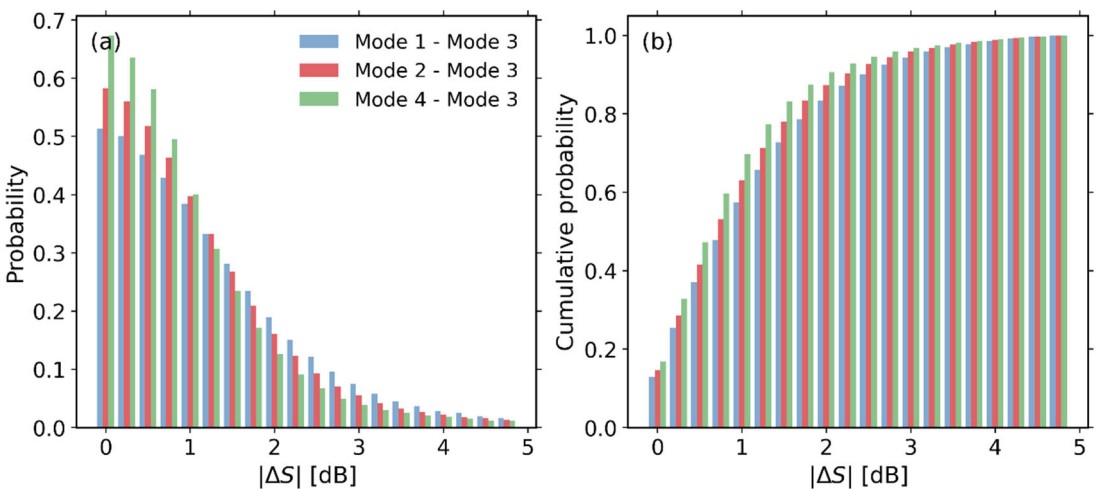


**Figure A1.** (a) Probability density and (b) cumulative distribution of spectral power ratio of meteorological signals between different modes at Ku-band.

## Appendix B: Determination of PDF$_{thresh}$

Here, we define PDF$_{thresh}$ = PDF$_{median}$ + $\alpha$PDF$_{SD}$. By varying $\alpha$, different values of PDF$_{thresh}$ can

be obtained. A similar quantitative evaluation can be made to find the appropriate value of $\alpha$ to maximize the sidelobe mitigation. Since the Doppler spectra observations at mode 3 for both radars are not affected by the sidelobe effect, they are used as the "reference data" at both Ku- and Ka-band. That is,

$$X_{Ku/Ka,ref} = X_{Ku/Ka,M3}^{mitigated} \qquad \text{(B1)}$$

Then, the standard deviation between spectral moments with different $\alpha$ and observations at mode 3 was calculated through Eq. (B1) and compared. Radar observations between 4.5 km and 6 km were evaluated, and the results for Ku- and Ka-band radars are given in Tab. B1 and B2, respectively. As we can see that the sidelobe artifacts has minimized impact on reflectivity and mean Doppler

velocity observations. After applying the PDF method, smaller standard deviation values can be
found for spectrum width, skewness and kurtosis. In addition, the performance of the PDF method
depends on the selection of α. The value of 1 seems to yield the best results. Smaller α (e.g., 0.2)
may mislabel sidelobe signals as meteorological echoes, while larger α (e.g., 1.8) may not able to
fully remove sidelobe signals. We have also tried other values such as 0.8 and 1.2 for α, and found
rather similar results with the use of 1. This demonstrates that α = 1 seems to be robust.

**Table B1.** Standard deviation (SD) for Ku-band spectral moments before and after the sidelobe
removal compared with observations at mode 3. Radar observations from 4.5 to 6 km are used for
comparison.

| Moments | Mode 2 | | | |
|---|---|---|---|---|
| | Before | After ($\alpha$=0.2) | After ($\alpha$=1) | After ($\alpha$=1.8) |
| Reflectivity (dBZ) | 1.08 | 1.08 | 1.08 | 1.08 |
| Mean velocity (m s$^{-1}$) | 0.13 | 0.14 | 0.14 | 0.14 |
| Spectrum width (m s$^{-1}$) | 0.13 | 0.06 | 0.06 | 0.06 |
| Skewness (-) | 2.88 | 0.47 | 0.40 | 0.49 |
| Kurtosis (-) | 31.38 | 6.08 | 5.40 | 5.61 |
| Moments | Mode 4 | | | |
| | Before | After ($\alpha$=0.2) | After ($\alpha$=1) | After ($\alpha$=1.8) |
| Reflectivity (dBZ) | 1.00 | 1.00 | 1.00 | 1.00 |
| Mean velocity (m s$^{-1}$) | 0.14 | 0.13 | 0.13 | 0.13 |
| Spectrum width (m s$^{-1}$) | 0.14 | 0.05 | 0.05 | 0.05 |
| Skewness (-) | 1.61 | 0.26 | 0.26 | 0.27 |
| Kurtosis (-) | 11.64 | 1.36 | 1.32 | 1.33 |

**Table B2.** Standard deviation (SD) for Ka-band spectral moments before and after the sidelobe
removal compared with observations at mode 3. Radar observations from 4.5 to 6 km are used for
comparison.

| Moments | Mode 2 | | | |
|---|---|---|---|---|
| | Before | After ($\alpha$=0.2) | After ($\alpha$=1) | After ($\alpha$=1.8) |
| Reflectivity (dBZ) | 0.90 | 0.90 | 0.90 | 0.90 |
| Mean velocity (m s$^{-1}$) | 0.13 | 0.13 | 0.13 | 0.13 |
| Spectrum width (m s$^{-1}$) | 0.20 | 0.07 | 0.07 | 0.07 |
| Skewness (-) | 3.29 | 1.00 | 0.56 | 0.56 |
| Kurtosis (-) | 39.66 | 13.95 | 6.57 | 6.51 |

| Moments | Mode 4 | | | |
|---|---|---|---|---|
| | Before | After ($\alpha$=0.2) | After ($\alpha$=1) | After ($\alpha$=1.8) |
| Reflectivity (dBZ) | 0.81 | 0.80 | 0.80 | 0.80 |
| Mean velocity (m s$^{-1}$) | 0.15 | 0.13 | 0.13 | 0.13 |
| Spectrum width (m s$^{-1}$) | 0.24 | 0.07 | 0.06 | 0.06 |
| Skewness (-) | 2.32 | 0.46 | 0.46 | 0.45 |
| Kurtosis (-) | 17.31 | 2.69 | 2.62 | 2.62 |

**Appendix C: Comparison of different range sidelobe mitigation methods**

This appendix shows the comparison of presented sidelobe mitigation algorithm (PDF method) with the threshold method (Liu and Zheng, 2019). The range sidelobe caused by pulse compression technology appears in both the upper and lower range gates of the target bin, which is weaker compared with the echo of the target. At Ku-band, the theoretical peak sidelobe ratio (the ratio of the main lobe peak power to the highest sidelobe peak power) is 36 dB and 30 dB for mode 2 and

mode 4, respectively. Figure C1 shows the comparison of sidelobe mitigation effects of the threshold method and the PDF method. The implementation of theoretical thresholds (Fig. C1a$_1$ and b$_1$) is insufficient to remove sidelobe signals. However, a smaller threshold may remove valid signals (Fig. C1a$_3$ and b$_3$). This effect is more evident in the zoomed-in plot (Fig. C2). In contrast, our algorithm is an adaptive method that efficiently removes sidelobe signals with the valid signal well preserved

(Fig. C1a$_4$ and b$_4$, Fig. C2a$_4$ and b$_4$).

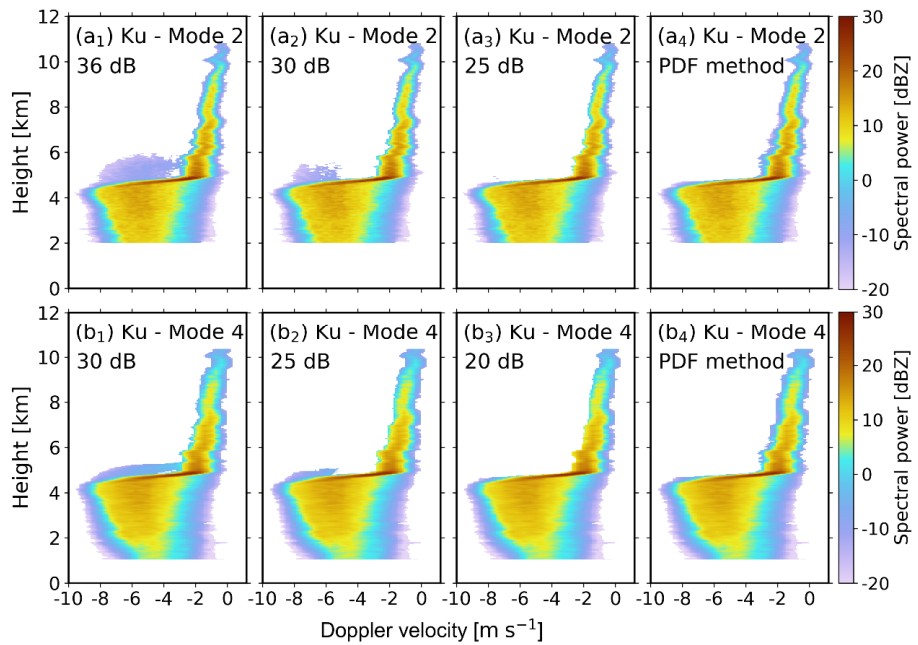

**Figure C1.** Ku-band Doppler power spectra after mitigating range sidelobe at modes 2 and 4 after

the sidelobe mitigation using the threshold method ($a_1$-$a_3$, $b_1$-$b_3$) and PDF method ($a_4$, $b_4$) on 6

June 2020 at 22:52:08 LST. The Doppler spectra observations before the sidelobe mitigation can

515                                  be found in Fig. 4$b_1$ and $d_1$.

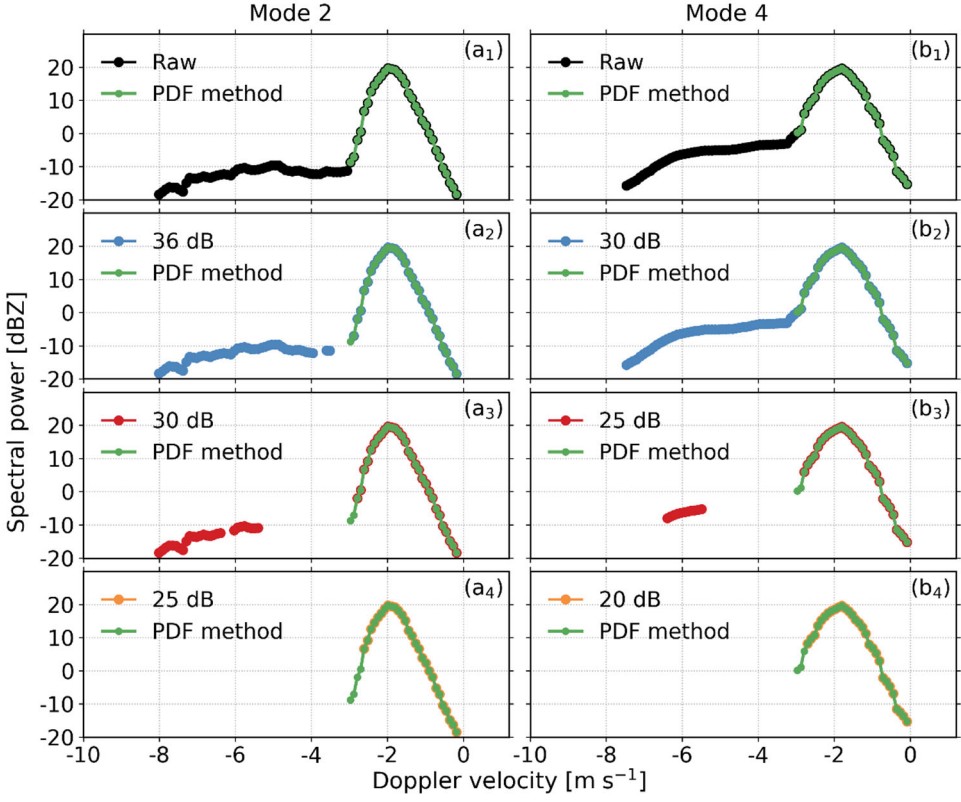

**Figure C2.** Ku-band Doppler power spectra recorded at (a) modes 2 and (b) 4 at 5.01 km after the

sidelobe mitigation using the threshold method and PDF method on 6 June 2020 at 22:52:08 LST.


**Appendix D: Statistical distribution of range sidelobe power as a function of height**

This appendix shows how much power is leaking into the range sidelobes. For a given velocity bin in a spectra profile, the maximum spectral power is denoted as $S_{peak}$, and the corresponding height is $H_{peak}$. Then, the spectral power of sidelobe is denoted as $S_{sidelobe}$ and the height as $H_{sidelobe}$.

The difference between $S_{peak}$ and $S_{sidelobe}$ and the corresponding $H_{peak}$ and $H_{sidelobe}$ were analyzed. As can be seen in Fig.C1, the pattern of range sidelobes at modes 2 and 4 are different due to their different pulse compression ratios.

The theoretical peak sidelobe ratio (the ratio of the main lobe peak power to the highest sidelobe peak power) depends on the transmitted waveform after pulse compression, and is 36 dB

and 30 dB for mode 2 and mode 4, respectively. Therefore, the sidelobe of mode 2 is weaker than that of mode 4. At the height close to $H_{peak}$, the power difference between $S_{sidelobe}$ and $S_{peak}$ can be much higher than the theoretical value due to the overlap of multiple sidelobes.

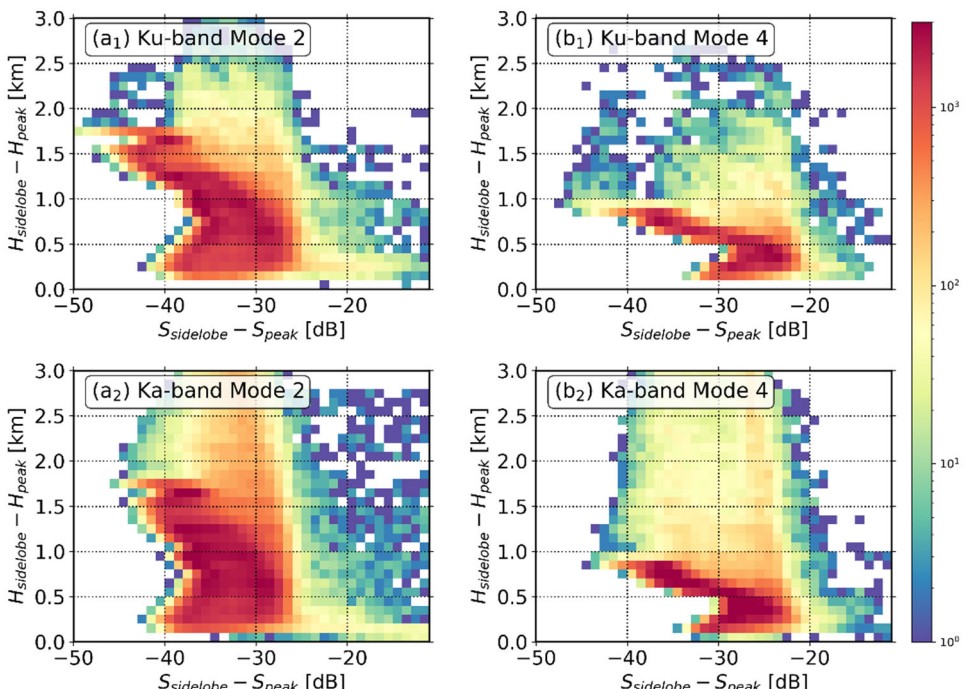

**Figure D1.** Statistical distribution of range sidelobe power as a function of heights at Ku-band

(upper panels) and Ka-band (lower panels).

*Author contributions.* HL and LL conceptualized this study; HD performed the investigation and co-wrote the draft with HL. All authors contributed to reviewing and editing the manuscript.

*Competing interests.* The authors declare that they have no conflict of interest.

*Financial support.* This research has been supported by the National Natural Science Foundation of China (grant no. 41875036, 42105141, U2142210), the Basic Research Fund of CAMS (451490), and the Postgraduate Research & Practice Innovation Program of Jiangsu Province (KYCX22_1153).

*Data availability.* The data used in this study can be made available upon request to the corresponding author.

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
