# Peer review of "Improved spectral processing for a multi-mode pulse compression Ka/Ku-band cloud radar system"

_Atmospheric Measurement Techniques, 2022_

## Referee Comment (RC1)

**Review of Improved spectral processing for a multi-mode pulse compression Ka/Ku-band cloud radar system**

by Han Ding, Haoran Li, Liping Liu

July 06, 2022

**1. General comments**

The authors present a framework for processing the Doppler spectra collected by a vertically-pointing, dual-frequency radar operating at the bands Ka and Ku. The framework includes a method for the removal of clutter and range sidelobe artifacts. The resulting clean spectra from three of the operational modes of the radar are merged and used to compute four moments of the spectrum.

In my opinion, the proposed framework is a good contribution to the radar community. It is presented in a clear way and most of the methods are described with a sufficient level of detail. However, there are few additional details (discussed in the next two sections of the review) that I believe need to be included in the explanation of the framework. I also found no major issues in the writing, but there are some instances (listed among the "Technical corrections") in which I suggest some modifications to make the text easier to understand.

I recommend the article for publication after addressing the issues listed below.

**2. Specific comments**

- Section 3.1
  The cutter mitigation algorithm is described clearly.
  The examples shown in the figures 1 and 3 illustrate its correct functioning when the tail of the power distribution of the precipitation signal is far enough from the 0 m/s fall velocity. In my opinion, it would be useful to briefly discuss in the text how the algorithm reacts in cases in which the meteorological signal is closer to 0 m/s (e.g. light precipitation, drizzle) and there is a more significant overlap between the signal and the clutter. Are there specific cases in which the algorithm fails or mislabels part of the precipitation as clutter?

- Sections 3.1 – 4.2
  The techniques illustrated in this section are always applied to the Ku-band in the various examples and their associated figures. This choice is motivated by the more common appearance of clutter for this band (Section 3.1) and the better functioning of the standard artifacts removal technique for the Ka-band (Section 3.2). Similarly, during the description of the mode merging in Section 4.1, only the statistics for $\Delta Z$ and $\Delta V$ for the Ku-band are shown.
  However, section 4.2 describes the result of the shift-then-average method for both the Ka and Ku bands. I have a couple of questions regarding this last section:

  - Are all the techniques described before section 4.2 also applied to the Ka-band?

  - If the same techniques developed for the Ku-band are used also for the Ka-band, are they applied in exactly the same way? (for example: is the mode 1 excluded from the shift-then-average method also for the Ka-band? Are the statistics of $\Delta Z$ and $\Delta V$ similar to the Ku-band case?

To address these two points in the main text, I suggest including a few sentences in Subsections 3.1, 3.2, and 4.1, mentioning how each step applies to the Ka-band and the (eventual) particularity of their implementation to this band.

**3. Technical corrections**

In each of the comments of this section, direct quotes from the article are highlighted in red and italic.

- Line 16
  "*Then, the abnormal distribution of the probability density of the Doppler spectrum in presence of range sidelobe due to the implementation of the pulse compression technique was identified and used to separate sidelobe artifacts.*"
  This sentence stands out to me as particularly long and slightly convoluted. If possible, I would suggest to re-phrase it, maybe splitting it into two shorter sentences to make it easier to understand for the reader.

- Line 28
  "*As a remote sensing instrument, cloud radars* [...]"
  Since "*cloud radars*" is plural, I would suggest using the plural for the first part of the sentence too (i.e. "As remote sensing instruments")

- Line 49
  "*Alternatively, cloud/precipitation signals can be well reserved if the clutter removal is made in the radar Doppler spectrum*"
  I did not fully understand the sentence. Is the term "*reserved*" correct here?

- Line 60
  "[...] *a wider pulse is used which on the other hand decreases* [...]"
  In my opinion, this part of the sentence would benefit from being re-phrased more clearly.

- Line 76
  "[...] *the emitting of long pulses leads to an increase in radar blind range,* [...]"
  In my opinion it should be "emission" instead of "*emitting*".

- Line 86
  Could you provide the altitude of the site at which the radar has been deployed?

- Lines 97-98
  "[...] *to improve the sensitivity to detect clouds with weaker radar echoes at higher latitudes*"
  Do you mean "higher altitudes"?

- Line 100
  "*There are four different modes routinely cycled in operations* [...]
  Since the four modes have already been introduced in the previous sentences, the beginning of this sentence feels like repetition. In my opinion, it could be rephrased as "These four different modes are routinely cycled in operations [...]".

- Line 105
  Since the height of the blind zone for two of the modes is mentioned, I would also explicitly write in the text the height for the remaining ones.

- Line 116
  "[...] *the implementation of pulse compression techniques in modes 2 and 4 usually results in significant range sidelobe around the melting layer*"
  Why specifically around the melting layer? Is it just because the melting layer is often characterized by a strong echo?

- Line 135
  "*The cause of such clutter signals is unclear yet and we hesitate to classify them to insects (Williams et al., (2018), since the spectral powers at different modes deviate from each other significantly.*"
  In my opinion, the verb "attribute" would fit better the sentence than "*classify*".
  Additionally, the parenthesis opened before the name "*Williams*" is not closed later in the phrase.

- Figure 1
  In my opinion, there is a mismatch between the label and the unit on the y-axis.
  The unit "*dBZ*" is used for reflectivity, but the label of the axis says "*power*".

- Line 172
  "*(see for example (Li and Moisseev, 2020))*"
  I think that the parenthesis around "*Li and Moisseev, 2020*" could be removed.

- Line 179
  "[...] *received spectral power from 2 km and 7 km*".
  From my understanding of the figure, in this sentence the "*and*" should be substituted by "to" (i.e. "from 2 km to 7 km").

- Line 180
  "*For Doppler spectra without the sidelobe contamination, PDFs are relatively uniform.*"
  From what I understood from panel (a) of figure 4, the term "*uniform*" may be confusing here. If I understood correctly, what is meant by this sentence is that the PDF observed at different range gates are similar to each other, in absence of the sidelobe issue. If this is the case, I suggest rewriting the sentence, avoiding the term "*uniform*", since it can create ambiguity with the idea of "uniform distribution" (which the various PDF of panel 4.a are not).

- Figure 4
  In my opinion, there are two small issues with this figure:

  - In panel (a) the orange and red curves cover completely the ones below. I would suggest adding some transparency to the lines so that also the curves below are visible. I also suggest inverting the order in which the lines are plotted, so that the light blue ones sit on top of the darker red ones.

  - Panel (b) is never mentioned in the main text of the article. Since the quantity $S_{thresh}$ is introduced in line 184 of the text, maybe you could expand the explanation including a mention of the panel (b).

- Line 186
  "*The procedures are briefly summarized as follows,*"
  I would replace the comma ("*,*") with a colon (":") since the procedure is provided in a numbered list just after this sentence.

- Lines 199-201
  "*Below half of the peak power above the noise level of the Doppler spectrum, find the power bins' probability density just exceeds the $PDF_{thresh}$, and the corresponding spectral power is set as $S_{thresh}$*"
  I did not fully understand this step in the procedure. Could the sentence be rewritten differently?

- Line 245
  "[...] *Doppler spectra observations from the modes 2, 3, and 4 were merged as follows* [...],"
  I suggest the same correction as for Line 186.

- Lines 270-273
  "*Although the agreement among different modes is better than that at Ku-band thanks to higher spectral velocity resolution and less uncertainties for the Ka-band radar, while the bias of kurtosis in snow at mode 3 (Fig. 11c) is more contrasting.*"
  I think that "Although" should be removed from the beginning of the sentence.

- Figures 12 and 13
  In the period of approximately 10 minutes before 21:00 LST, in both bands it is possible to see some very faint returns around 2 km of altitude. Do you know what is causing their appearance? In case it is unfiltered clutter, I would recommend discussing its appearance in the text, describing briefly why the proposed method does not filter it.

- Lines 314-315
  "[...] *and the results show good performance of clutter/sidelobe suppression and spectral merging.*"
  Since the performances were not measured quantitatively, I would modify this sentence by using a less strong statement (e.g. "a visual inspection of the processed data suggests that clutter/sidelobe suppression and spectral merging demonstrated good performances").

---

## Referee Comment (RC2)

**Review for the manuscript "Improved spectral processing for a multi-mode pulse compression Ka/Ku-band cloud radar system" by Ding et al.**

The manuscript describes a set of processing techniques applicable to spectral observations from cloud radars. The techniques include clutter filtering, mitigation of artefacts resulting from pulse compression, and merging of observations taken with different pulse modes.

The manuscript has many flaws, to name few:

1. The authors claim "…the results show good performance of clutter/sidelobe suppression and spectral merging", but the manuscript completely lacks any quantitative evaluation of the proposed methods.

2. The described methods contain a number of decision rules (e.g. Figs. 2 and 6). Intuitively, these rules may sound to be reasonable. I, however, would certainly expect a statistical justification of the used rules. For instance, the authors write "The selection of deltaS=3dB is a compromise between the radars' observation uncertainty and the spectral ratio between different observing modes." I do not think this is enough. First, observational uncertainties depend on the operational settings. What if I use different settings on my radar, shall I change the settings to the ones used for the manuscript in order to apply the method? Or how shall I adapt the method to use it with different settings? Second, which rule was used to get the compromise? What I, as a radar operator, would like to see is, for a given pulse modes, what is the probability not to filter clutter? What is the probability to erroneously filter out a spectral line with meteorological signal only? How these probabilities depend on the pulse modes? How would these probabilities change if different thresholds are used? There are no answers in the manuscript.

3. All methods are illustrated using rain cases. How would these methods work under other conditions? For example, would the clutter-filtering algorithm still be able to discriminate between a thin liquid layer cloud with highly variable reflectivity and non-meteorological targets? I would expect that the performance in the statistical sense changes and I want to know how, before I apply the proposed method. Also, it is hard to say how well the side-lobe mitigation algorithm would perform in solid precipitation because there will be no clear separation between the sidelobe and meteorological signal as in case of rain right above the melting layer.

4. In the case of a novel processing technique with a number of subjectively chosen parameters, I would at least expect a comparison to a reference radar which does not have artefacts in measurements (e.g. a magnetron-based cloud radar).

5. Since the authors mention that there are alternative techniques available to mitigate side-lobes. I would also expect a comparison of the proposed methods with the available one.

Based already on these points I conclude that the manuscript is not ready for a further more detailed review and I recommend rejection.

---

## Author Comment (AC1)

The authors present a framework for processing the Doppler spectra collected by a vertically-pointing, dual frequency radar operating at the bands Ka and Ku. The framework includes a method for the removal of clutter and range sidelobe artifacts. The resulting clean spectra from three of the operational modes of the radar are merged and used to compute four moments of the spectrum.

In my opinion, the proposed framework is a good contribution to the radar community. It is presented in a clear way and most of the methods are described with a sufficient level of detail. However, there are few additional details (discussed in the next two sections of the review) that I believe need to be included in the explanation of the framework. I also found no major issues in the writing, but there are some instances (listed among the "Technical corrections") in which I suggest some modifications to make the text easier to understand.

I recommend the article for publication after addressing the issues listed below.

We sincerely appreciate the reviewer for the positive comments on our paper. We have amended the manuscript as suggested. Please see below our response to your comments.

**Specific comments**
- Section 3.1
  The clutter mitigation algorithm is described clearly. The examples shown in the figures 1 and 3 illustrate its correct functioning when the tail of the power distribution of the precipitation signal is far enough from the 0 m/s fall velocity. In my opinion, it would be useful to briefly discuss in the text how the algorithm reacts in cases in which the meteorological signal is closer to 0 m/s (e.g. light precipitation, drizzle) and there is a more significant overlap between the signal and the clutter. Are there specific cases in which the algorithm fails or mislabels part of the precipitation as clutter?

  We thank the reviewer for this question which led to a more thorough discussion on our method.

  The declutter method in this paper deals with clutter signals that are not consistent at different observing modes (See Fig.2 in the revised manuscript). In contrast, the meteorological signals are usually coherent and are consistent at different observing modes. If meteorological signals coexist with clutter signals, there are two scenarios:
  1) If the spectral powers of meteorological signals exceed those of clutter signals, the meteorological signals will be preserved since they are consistent and $|\Delta S|$ is not expected to be large.
  2) If the spectral powers of meteorological signals are below those of clutter signals, clutter signals will dominate the spectral power. If they are consistent at different modes ($|\Delta S|$ is not very large), they can be mislabeled as

precipitation/clouds.

The critical point here is to get a threshold for $|\Delta S|$. As we know, the selection of the threshold is a comprise between false-alarm and miss hit. We want to preserve the meteorological signals at our best, therefore we checked that for the meteorological signals how large the $|\Delta S|$ can be. We attach the statistical plots of meteorological signals (Doppler velocity of 2 ~ 5 m/s) below (Appendix A in the revised manuscript). We have used 3 dB since the probability of $|\Delta S|$ tends to be flat after this value. If a value higher than 3 dB is used, more clutter signals can be mislabeled as precipitation.

[Figure]

(a) Probability density and (b) cumulative distribution of spectral power ratio of meteorological signals between different modes at Ku-band

In the revised manuscript, we have clarified this point in Section 3.1.

"The selection of the threshold is a comprise between false-alarm and miss hit. We want to preserve the meteorological signals at our best, therefore we checked the magnitudes of $|\Delta S|$ for meteorological signals. Figure A1 (Appendix) presents the statistical plot of $|\Delta S|$ for meteorological signals (height of 2 ~ 3 km and Doppler velocity of 2 ~ 5 m s$^{-1}$). It appears that the probability of $|\Delta S|$ tends to be flat after 3 dB, and the use of 3 dB can ensure that 95.6 % of precipitation cases are well preserved (Fig. A1). Therefore, 3 dB is used in this study. If a larger threshold is employed, we expect more clutter signals will be mislabeled as precipitation."

The clutter removal method was modified in the revised manuscript (see Fig. 3), we have added more discussions as below:

"It should be noted that this method relies on observations recorded at different observing modes. However, the sensitivities of different modes are not identical. Therefore, if the clutter is presented in the most sensitive mode (e.g., mode 2) only, it cannot be filtered out with the $|\Delta S|$ method. In this case, the width of valid meteorological spectral mode is assumed to be longer than 2 m s$^{-1}$, otherwise it is attributed to clutter. We are aware that Shupe et al (2004) have used a width of

0.448 m s$^{-1}$ to identify supercooled liquid water. We have tried this value, but the width of clutter present in this dual-wavelength radar system easily exceeds 1 m s$^{-1}$ (Fig. 2). Actually, the selection of the spectrum width is similar with the use of a signal-to-noise ratio (SNR) value in noise-removal. Higher SNR means a stricter noise-removal but higher chance of losing valid signals. We have tested the width of 1, 1.5, 2, and 3 m s$^{-1}$ (visual inspection, not shown), and found that 2 m s$^{-1}$ can effectively remove clutter signals for both radars though very light precipitation (detected by the most sensitive mode only) can be removed as well. Admitting this potential issue, it suffices the application in rainfall. In addition, for clouds with highly variable reflectivity, the presented algorithm may mislabel them as clutter according to our assumption that meteorological signals are coherent in a round of observation (28s)."

- Section 3.1 – 4.2
  The techniques illustrated in this section are always applied to the Ku-band in the various examples and their associated figures. This choice is motivated by the more common appearance of clutter for this band (Section 3.1) and the better functioning of the standard artifacts removal technique for the Ka-band (Section 3.2). Similarly, during the description of the mode merging in Section 4.1, only the statistics for $\Delta Z$ and $\Delta V$ for the Ku-band are shown. However, section 4.2 describes the result of the shift-then-average method for both the Ka and Ku bands. I have a couple of questions regarding this last section:
  - Are all the techniques described before section 4.2 also applied to the Ka-band?
  - If the same techniques developed for the Ku-band are used also for the Ka-band, are they applied in exactly the same way? (for example: is the mode 1 excluded from the shift-then-average method also for the Ka-band? Are the statistics of $\Delta Z$ and $\Delta V$ similar to the Ku-band case?

  Sorry for the unclear description in the original manuscript. The techniques were applied to Ka-band. The different point is that Ka-band observations at modes 3 and 4 were used, while Ku-band uses data from modes 2, 3, and 4. In the revised manuscript, the procedures for generating the estimates of spectral moments are presented in Fig. 14.

  In addition, $\Delta Z$ and $\Delta V$ plot of Ka-band is given in Fig. 10. We can see that the coherent integration has a decent impact on Ka-band, therefore we did not use mode 2 data at Ka-band.

**Technical comments**
  - Line 16

    "Then, the abnormal distribution of the probability density of the Doppler

    spectrum in presence of range sidelobe due to the implementation of the pulse compression technique was identified and used to separate sidelobe artifacts."

This sentence stands out to me as particularly long and slightly convoluted. If possible, I would suggest to re-phrase it, maybe splitting it into two shorter sentences to make it easier to understand for the reader.

This sentence has been rewritten as "Then, for the Doppler spectrum affected by the range sidelobe due to the implementation of the pulse compression technique, the characteristics of the probability density distribution of the spectral power were used to identify the sidelobe artifacts."

- Line 28

  "As a remote sensing instrument, cloud radars [...]"

  Since "cloud radars" is plural, I would suggest using the plural for the first part of the sentence too (i.e. "As remote sensing instruments")

  Corrected.

- Line 49

  "Alternatively, cloud/precipitation signals can be well reserved if the clutter

  removal is made in the radar Doppler spectrum"
  I did not fully understand the sentence. Is the term "reserved" correct here?

  This sentence has been revised to "Alternatively, cloud/precipitation signals can be discriminated from clutter properly if the clutter removal is made in the radar Doppler spectrum".

- Line 60

  "[...] a wider pulse is used which on the other hand decreases [...]"

  In my opinion, this part of the sentence would benefit from being re-phrased more clearly.

  The sentence has been rewritten to "To enhance the detection sensitivity, modulated wide pulses are transmitted and then compressed into short pulses after received."

- Line 76

  "[...] the emitting of long pulses leads to an increase in radar blind range, [...]"

  In my opinion it should be "emission" instead of "emitting".

  Corrected.

- Line 86

  Could you provide the altitude of the site at which the radar has been deployed?

  The altitude of the site where the radar has been deployed is 80.3 m. We have added the altitude to the revised manuscript.

- Lines 97-98

  "[...] to improve the sensitivity to detect clouds with weaker radar echoes at higher latitudes"

  Do you mean "higher altitudes"?

  Yes, we have corrected it in the revised manuscript.

- Line 100

  "There are four different modes routinely cycled in operations [...]

  Since the four modes have already been introduced in the previous sentences, the beginning of this sentence feels like repetition. In my opinion, it could be rephrased as "These four different modes are routinely cycled in operations [...]".

  We have rephrased this sentence to "These four different modes are routinely cycled in operations and each mode takes 7 s to finish the observation.".

- Line 105

  Since the height of the blind zone for two of the modes is mentioned, I would also explicitly write in the text the height for the remaining ones.

  Thanks for the suggestion. We have added the blind zone of the other two modes in the revised manuscript. "The blind zones of modes 1 and 3 are 30 m."

- Line 116

  "[...] the implementation of pulse compression techniques in modes 2 and 4 usually results in significant range sidelobe around the melting layer"

  Why specifically around the melting layer? Is it just because the melting layer is often characterized by a strong echo?

  The range sidelobe caused by pulse compression technology is present in both the upper and lower range gates of the target bin, which is weak compared with the echo of the target. The theoretical peak sidelobe ratio (the ratio of the main lobe peak power to the highest sidelobe peak power) is 36 dB and 30 dB for mode 2 and mode 4, respectively. Our statistics (Fig. D1) show that the sidelobe signals are usually below -20 dB. Since the reflectivity enhancement in the melting layer usually do not exceed 10 dB (Li et al., 2020), the sidelobe

contamination in rain is not significant. However, the fall velocity of snow is much slower than rain drops. Namely, no meteorological signals present in the range of 3 ~ 10 m/s and the sidelobe signal becomes evident.

- Line 135

"The cause of such clutter signals is unclear yet and we hesitate to classify

them to insects (Williams et al., (2018), since the spectral powers at different modes deviate from each other significantly."
In my opinion, the verb "attribute" would fit better the sentence than "classify". Additionally, the parenthesis opened before the name "Williams" is not closed later in the phrase.
Thank the reviewer very much for the suggestion. The verb "classify" has been changed to "attribute" and the parenthesis has been closed.

- Figure 1
In my opinion, there is a mismatch between the label and the unit on the y-axis. The unit "dBZ" is used for reflectivity, but the label of the axis says "power".

The unit of Doppler power spectral density data is $\text{mm}^6 \, \text{m}^{-3} \, (\text{m s}^{-1})^{-1}$. There is no accepted nomenclature for denoting the spectral power in the dB scale according to Li and Moisseev (2020), so we are going to with the unit "dBZ", but we added some explanation in the caption of the figure.

- Line 172

"(see for example (Li and Moisseev, 2020))"

I think that the parenthesis around "Li and Moisseev, 2020" could be removed.
The parenthesis around "Li and Moisseev, 2020" has been removed.

- Line 179

"[...] received spectral power from 2 km and 7 km".

From my understanding of the figure, in this sentence the "and" should be

substituted by "to" (i.e. "from 2 km to 7 km").

Corrected.

- Line 180

"For Doppler spectra without the sidelobe contamination, PDFs are relatively

uniform."
From what I understood from panel (a) of figure 4, the term "uniform" may be

confusing here. If I understood correctly, what is meant by this sentence is that the PDF observed at different range gates are similar to each other, in absence of the sidelobe issue. If this is the case, I suggest rewriting the sentence, avoiding the term "uniform", since it can create ambiguity with the idea of "uniform distribution" (which the various PDF of panel 4.a are not).

We have rewritten the description of figure 4, please check it in the response below.

- Figure 4
  In my opinion, there are two small issues with this figure:
  ◦ In panel (a) the orange and red curves cover completely the ones below. I would suggest adding some transparency to the lines so that also the curves below are visible. I also suggest inverting the order in which the lines are plotted, so that the light blue ones sit on top of the darker red ones.
  ◦ Panel (b) is never mentioned in the main text of the article. Since the quantity $S_{thresh}$ is introduced in line 184 of the text, maybe you could expand the explanation including a mention of the panel (b).

To present the results to the reader more clearly, we modified Fig. 4 in the original manuscript to show only the probability distributions of spectral power of range bins at 2.4 km, 5.01 km, and 6.6 km, which respectively represent the liquid precipitation, Doppler spectra contaminated by range sidelobe, and solid precipitation (Fig. 5 in the revised manuscript). The following descriptions have been added to the revised manuscript:

"An interesting feature of the range sidelobe caused by pulse compression is that its spectral power is much flatter than cloud and precipitation signals. Figure 5a shows the probability density functions (PDFs) of received spectral power at 2.4 km, 5.01 km, and 6.6 km, which respectively represent the liquid precipitation, Doppler spectrum contaminated by range sidelobe, and solid precipitation. For the sidelobe-contaminated Doppler spectrum, It can be seen that the range bins contaminated by range sidelobe have different spectral power distributions, the peak of the PDFs appears close to the noise level and is mostly below 15 dB above the noise level. A closer look into the radar Doppler spectra at 5.01 km (Fig. 6a) shows that the strong PDF peak in Fig. 5b is explained by the relatively flat range sidelobe signals. Here, we introduce a parameter spectral power threshold ($S_{thresh}$) to distinguish the range sidelobe from meteorological signals."

[Figure]

(a): Probability distributions of Doppler spectra at 2.4 km (liquid precipitation), 5.01 km (melting layer), and 6.6 km (solid precipitation) at mode 2; (b): Probability distribution of Doppler spectrum recorded at 5.01 km.

- Line 186

"The procedures are briefly summarized as follows,"

I would replace the comma (",") with a colon (":") since the procedure is provided in a numbered list just after this sentence.
Corrected.

- Lines 199-201

"Below half of the peak power above the noise level of the Doppler spectrum, find the power bins' probability density just exceeds the $PDF_{thresh}$, and the corresponding spectral power is set as $S_{thresh}$"
I did not fully understand this step in the procedure. Could the sentence be rewritten differently?
This sentence has been revised to:

"1) Sort the spectral power values above noise level in an ascending order to get a PDF curve of each Doppler spectrum;
2) Calculate the median and standard deviation (SD) of the PDFs, set $PDF_{thresh} = PDF_{median} + PDF_{SD}$; Note that the determination of this relation is given in Appendix B.
3) Below half of the peak power above the noise level of the Doppler spectrum, find the power bins' probability density just exceeds the $PDF_{thresh}$, and the corresponding spectral power is set as $S_{thresh}$; (The range of $PDF_{thresh}$ is limited to half of the peak power above the noise level to avoid finding the $PDF_{peak}$ corresponding to large spectral power, which makes the determined $S_{thresh}$ corresponds well to the power of sidelobe in this way.)

4) If the spectrum power with the Doppler velocity larger than the mean Doppler velocity is below the $S_{thresh}$, then it is flagged as sidelobe."

- Line 245

"[...] Doppler spectra observations from the modes 2, 3, and 4 were merged as

follows [...],"
I suggest the same correction as for Line 186.
Corrected.

- Lines 270-273

"Although the agreement among different modes is better than that at Ku-band

thanks to higher spectral velocity resolution and less uncertainties for the Ka-band radar, while the bias of kurtosis in snow at mode 3 (Fig. 11c) is more contrasting."
I think that "Although" should be removed from the beginning of the sentence.
Corrected.

- Figures 12 and 13
In the period of approximately 10 minutes before 21:00 LST, in both bands it is possible to see some very faint returns around 2 km of altitude. Do you know what is causing their appearance? In case it is unfiltered clutter, I would recommend discussing its appearance in the text, describing briefly why the proposed method does not filter it.
We checked why the clutter was still there. This is due to the different sensitivities of different observing modes. If the clutter signal is detected only by the most sensitive mode, then our method will fail to filter it out because there are no signals from other modes to compare it with. As shown in Fig.12 (Fig. 15 in the revised manuscript), the clutters that are not filtered out exist above 2 km which is because the blind zones of the most sensitive mode (mode 2) of our radar are below 2 km. To clarify this, these sentences have been added to the revised manuscript.

"It should be noted that this method relies on observations at different observing modes. However, the sensitivities of different modes are not identical. Therefore, if the clutter is presented in only one mode, it cannot be filtered out. " has been added in Section 3.1.

"It can be seen in Fig. 15 that there are still clutter signals above 2 km at Ku-band between 20:45 and 21:00 LST, which are all only detected by the most sensitive mode (mode 2). The clutter was not filtered out because no signals were detected by other modes." has been added in Section 5.1.

- Lines 314-315

"[...] and the results show good performance of clutter/sidelobe suppression and spectral merging."
Since the performances were not measured quantitatively, I would modify this sentence by using a less strong statement (e.g. "a visual inspection of the processed data suggests that clutter/sidelobe suppression and spectral merging demonstrated good performances").

In the revised manuscript, we have added the quantitative evaluation. This statement has been changed to "and the quantitative evaluations of the processed data suggests that clutter/sidelobe suppression and spectral merging results demonstrated good performance."

---

## Author Comment (AC2)

The manuscript describes a set of processing techniques applicable to spectral observations from cloud radars. The techniques include clutter filtering, mitigation of artefacts resulting from pulse compression, and merging of observations taken with different pulse modes.

We sincerely appreciate the reviewer for pointing out the aspects which were not well addressed in the original submission. We have made major revision to the manuscript based on your comments. Please see below our response to your comments.

The manuscript has many flaws, to name few:
1. The authors claim "…the results show good performance of clutter/sidelobe suppression and spectral merging", but the manuscript completely lacks any quantitative evaluation of the proposed methods.

We agree with the reviewer that a quantitative evaluation would definitely strengthen our conclusions. In the revised manuscript, we have added a section "5.3 Quantitative evaluation" which presents quantitative comparison of spectral moments before and after the spectral processing. The evaluation of clutter removal and sidelobe mitigation are given in sections "5.3.1" and "5.3.2", respectively. Please see the revision for details.

2. The described methods contain a number of decision rules (e.g. Figs. 2 and 6). Intuitively, these rules may sound to be reasonable. I, however, would certainly expect a statistical justification of the used rules. For instance, the authors write "The selection of deltaS=3dB is a compromise between the radars' observation uncertainty and the spectral ratio between different observing modes." I do not think this is enough. First, observational uncertainties depend on the operational settings. What if I use different settings on my radar, shall I change the settings to the ones used for the manuscript in order to apply the method? Or how shall I adapt the method to use it with different settings? Second, which rule was used to get the compromise? What I, as a radar operator, would like to see is, for a given pulse modes, what is the probability not to filter clutter? What is the probability to erroneously filter out a spectral line with meteorological signal only? How these probabilities depend on the pulse modes? How would these probabilities change if different thresholds are used? There are no answers in the manuscript.

We feel sorry that the process of threshold selection was missed in the original manuscript. In the revised manuscript, the details are given below,

1.  The selection of $|\Delta S| = 3$ dB in Fig. 3 (revised manuscript).

    In the revised manuscript, we have clarified this point in Section 3.1:

    "The selection of the threshold is a comprise between false-alarm and miss hit. We

want to preserve the meteorological signals at our best, therefore we checked the magnitudes of $|\Delta S|$ for meteorological signals. Figure A1 (Appendix) presents the statistical plot of $|\Delta S|$ for meteorological signals (height of 2 ~ 3 km and Doppler velocity of 2 ~ 5 m s$^{-1}$). It appears that the probability of $|\Delta S|$ tends to be flat after 3 dB, and the use of 3 dB can ensure that 95.6 % of precipitation cases are well preserved (Figure A1). Therefore, 3 dB is used in this study. If a larger threshold is employed, we expect more clutter signals will be mislabeled as precipitation."

2. $PDF_{thresh} = PDF_{median} + PDF_{SD}$ in Fig. 7 (revised manuscript).

In the revised manuscript, we have clarified this point in Appendix B. By varying different α in $PDF_{thresh} = PDF_{median} + \alpha PDF_{SD}$, we show that a value around 1 is a reasonable value for α. Please see the details in the revised manuscript.

3. All methods are illustrated using rain cases. How would these methods work under other conditions? For example, would the clutter-filtering algorithm still be able to discriminate between a thin liquid layer cloud with highly variable reflectivity and non-meteorological targets? I would expect that the performance in the statistical sense changes and I want to know how, before I apply the proposed method. Also, it is hard to say how well the side-lobe mitigation algorithm would perform in solid precipitation because there will be no clear separation between the sidelobe and meteorological signal as in case of rain right above the melting layer.

This Ku/Ka-band radar system is deployed in Southern China, and no snowfall observations have been recorded yet. Therefore, we cannot give an assessment on other conditions. In the revised manuscript, we have stated this point in the Summary.

"The presented methods mainly deal with the challenges in observing stratiform rainfall events in Southern China, given the weaker signal attenuation at both bands compared with that in convective precipitation. We are aware that cloud radars have proven to be an effective tool for snowfall observations (e.g., Kollias et al., 2007; Li et al., 2021), the applicability of the presented framework in snowfall is yet clear despite that the sidelobe contamination in snowfall is not as significant as that in the presence of melting layer."

Regarding the applicability of the clutter removal method in clouds with highly variable reflectivity. We have added the following discussions in Section 3.1.

"In addition, for clouds with highly variable reflectivity, the presented algorithm may mislabel them as clutter according to our assumption that meteorological signals are coherent in a round of observation (28s)."

4. In the case of a novel processing technique with a number of subjectively chosen parameters, I would at least expect a comparison to a reference radar which does not

have artefacts in measurements (e.g. a magnetron-based cloud radar).

We do not have magnetron-based cloud radars working at the observation station, but there is a collocated C-band frequency modulated continuous wave radar (FMCW) radar. The C-band radar's data products include reflectivity, Doppler velocity, and spectrum width. In the revised manuscript, we have added the comparison of spectrum width with C-band radar observations regarding the effect of sidelobe mitigation. Please see details in Section "5.2 Comparison with a C-band radar".

5. Since the authors mention that there are alternative techniques available to mitigate side-lobes. I would also expect a comparison of the proposed methods with the available one.
Another sidelobe mitigation method is based on the theoretical power of the sidelobe to set the threshold for sidelobe identification. If the echo of the main lobe is detected correctly, then we can calculate the theoretical sidelobe power by knowing the peak sidelobe ratio. But in fact, the precipitation echoes are not isolated targets, a range gate can receive range sidelobes from several other gates at the same time. Therefore, it is difficult to remove all the sidelobes with a fixed threshold, which sometimes requires many rounds of operation.

In the revised manuscript, we have added the comparison to the threshold method, please see details in the last paragraph of Section 3.2.

---

## Author Comment (AC3)

This manuscript presents methods to remove clutter and range sidelobe artifacts in vertically pointing radar Doppler velocity power spectra. The manuscript also describes a method to combine spectra from four different modes to produce a merged moment dataset.

It appears that the authors have done some good research exploring Doppler velocity power spectra, yet, the manuscript does not provide enough quantitative analysis for an AMT reader to adapt the proposed methodologies to different radar systems.

Also, the manuscript does not provide enough examples of "clutter" in different weather conditions to convince this reviewer that the radar is observing non-meteorological clutter. From the imagery presented in the manuscript, it appears to this reviewer that

the "clutter" is actually clouds being detected in the same radar resolution volume as

precipitation. One possibility is instead of identifying and removing "clutter" from the power spectra, the manuscript should explore using multiple-peak processing methods to identify multiple hydrometeor populations occurring within the same range resolution volume. For example, in Fig. 1a, it appears to this reviewer that the radar is detecting both raindrops (with Doppler velocities exceeding 8 m/s) and cloud particles (with Doppler velocities near zero).

I encourage the authors to continue their work analyzing Doppler velocity spectra and make improvements to this manuscript.

The authors would like to thank the reviewer for constructive comments on the manuscript. The comments will help to sharpen and clarify the paper, all of them will be addressed in some manner. Please see the point-to-point response below in blue color.

**Major concerns**
- 1. Line 117.
  The text states, "…the implementation of pulse compression techniques in modes 2 and 4 usually results in significant range sidelobe around the melting layer, which does not significantly affect Ze and V estimates, but can severely degrade the estimation of spectrum width." This sentence is not logical. If there is "significant" range sidelobes, then power that should be assigned to the central range gate is being "significantly" distributed into different range gates, and Ze and V will be incorrect in the central range gate and in the range sidelobe gates. Will the error in Z and V be "significant"? This manuscript needs to describe how much power is leaking into the range sidelobes and how that is affecting the Ze and V estimates. Including that quantitative analysis through simulations or detailed analysis would be valuable to AMT readers.

  We agree with the reviewer. In the revised manuscript, we have added a new section "5.3 Quantitative evaluation". In section "5.3.2 Sidelobe mitigation", we have

quantitatively analyzed the impact of sidelobe removal on spectral moment estimation (Tab. B1 and B2).

Regarding the magnitude of power leakage, we have added a new paragraph in Section "5.1 Case study".

"In addition, we have calculated statistics of the power leakage to range sidelobe, and the results for Ku-/Ka-band radars are given in Fig. D1 (Appendix). The results show that the sidelobe signals are usually below -20 dB. Since the reflectivity enhancement in the melting layer usually does not exceed 10 dB (Li et al., 2020), the sidelobe contamination in rain is not significant. However, the fall velocity of snow is much slower than rain drops. Namely, no meteorological signals present in the range of $3 \sim 10$ m s$^{-1}$ and the sidelobe signal becomes evident."

- 2. Figure 1 and Line 135.
  "The cause of such clutter signals is unclear yet we hesitate to classify them to insects (Williams et al. 2018), since the spectral powers at different modes deviate from each other significantly." This reviewer agrees, the spectral peaks near zero velocities are probably not due to scattering from insects. From examining Fig. 12, it appears that the spectral power near zero velocity is scattering from clouds. Thus, Fig. 1a probably shows return power from precipitation and cloud particles. In Fig. 12, between time 20:00 through 21:00 LST and near 2 km, there is a cloud feature in the reflectivity and velocity time-height cross-sections that suggest this 'clutter' peak near zero velocity in Fig. 1a is scattering from cloud particles being detected in the more sensitive modes 4 and 2. The manuscript should provide more imagery of this "clutter" signal to convince the AMT reader that the radar is observing non-meteorological clutter. For example, a time-spectra plot showing spectra at one range gate over multiple profiles could indicate whether this signal is coherent over many profiles, indicative of a cloud.

  We thank the reviewer for this good suggestion. In the revised manuscript, we have added a time-spectra plot and the following descriptions:

  "Figure 2 shows the time series of Doppler velocity spectra on 6 June 2020 from 22:40 to 23:01 LST at 2.34 km range (the same range bin as Fig. 1). The clutter signals are in the vicinity of 0 m s$^{-1}$ and are not continuous with time. Compared with meteorological signals, it appears that clutter echoes randomly occur with some dependence on the observing mode."

  Regarding the clutter in Fig. 12 (Fig. 15 in the revised manuscript), we have checked the spectral observations and found that the unsuccessful declutter was because no signals from another mode can be used to compare with the most sensitive mode (mode 2). This can also be evidenced in Fig. 2 in the revised manuscript. Clutter signals at 22:48 presented at mode 2 (the most sensitive mode), and our algorithm cannot identify them. We have added the following discussions in the last paragraph of section 3.1

"It should be noted that this method relies on observations recorded at different observing modes. However, the sensitivities of different modes are not identical. Therefore, if the clutter is presented in the most sensitive mode (e.g., mode 2) only, it cannot be filtered out with the $|\Delta S|$ method. In this case, the width of valid meteorological spectral mode is assumed to be longer than 2 m s$^{-1}$, otherwise it is attributed to clutter. We are aware that Shupe et al (2004) have used a width of 0.448 m s$^{-1}$ to identify supercooled liquid water. We have tried this value, but the width of clutter present in this dual-wavelength radar system easily exceeds 1 m s$^{-1}$ (Fig. 2). Actually, the selection of the spectrum width is similar with the use of a signal-to-noise ratio (SNR) value in noise-removal. Higher SNR means a stricter noise-removal but higher chance of losing valid signals. We have tested the width of 1, 1.5, 2, and 3 m s$^{-1}$ (visual inspection, not shown), and found that 2 m s$^{-1}$ can effectively remove clutter signals though very light precipitation (detected by the most sensitive mode only) can be removed as well. Admitting this potential issue, it suffices the application in rainfall. In addition, for clouds with highly variable reflectivity, the presented algorithm may mislabel them as clutter according to our assumption that meteorological signals are coherent in a round of observation (28s)."

- 3. Line 144.

  "The selection of |Delta S| = 3 dB is a compromise …". The manuscript has omitted important processing steps needed to compare different operating modes collected from the same radar system. Specifically, the manuscript needs to describe how the spectra from the different modes are cross-calibrated. Each mode has their own noise level, sensitivity, and velocity resolution such that the Doppler velocity power spectral density magnitudes will not be the same for the different modes. The modes must be cross-calibrated and scaled in order to produce the spectra shown in Fig. 1 in units of dBZ (this includes a calibration offset plus range squared correction). Regarding the 3 dB threshold, the manuscript needs more analysis showing the range of power differences between the modes in order to justify a particular threshold so that AMT readers to apply the proposed technique to other radars.

  We agree with the reviewer. In the revised manuscript, we have clarified this point in Section 3.1:

  "The selection of the threshold is a comprise between false-alarm and miss hit. We want to preserve the meteorological signals at our best, therefore we checked the magnitudes of $|\Delta S|$ for meteorological signals. Figure A1 (Appendix) presents the statistical plot of $|\Delta S|$ for meteorological signals (height of 2 ~ 3 km and Doppler velocity of 2 ~ 5 m s$^{-1}$). It appears that the probability of $|\Delta S|$ tends to be flat after 3 dB, and the use of 3 dB can ensure that 95.6 % of precipitation cases are well preserved (Figure A1). Therefore, 3 dB is used in this study. If a larger threshold is employed, we expect more clutter signals will be mislabeled as precipitation."

4. Section 3.2.

This section is very confusing for the reader and needs to be re-written.

- a. The manuscript needs text describing the physical process that is causing the range sidelobes. That is, the de-coding of the phase modulated signal has errors and is causing power to appear in the wrong range gates.

Our Ka/Ku-band cloud radar use the linear frequency modulation (LFM) technique for signal modulation, the range sidelobe is generated by the response of the signal outside the matched filter. We have added the description of the cause of the range sidelobe.

"To improve both the radar detection performance and range resolution, Linear Frequency Modulation was used to widen the signal bandwidth when transmitting pulses in modes 2 and 4 at both Ka- and Ku-band. But, the matched pulse compression filter output exhibits sidelobe behavior, making the power of range sidelobe appear in the wrong range gates." has been added to Section 3.2 in the revised manuscript.

- b. Line 173.

The description "sidelobe caused by the pulse compression drags a long tail in the relatively large velocity bins…" is not correct or is poorly worded. The range sidelobe does not move power (or drag power) into different velocity bins to cause a long tail. The power appears in different range gates at the same Doppler velocity. Please clarify text.

We have rewritten the sentence to clarify the description.

"Compared with radar Doppler spectrum observations without the sidelobe contamination (see for example Li and Moisseev, 2020), Doppler spectra above the melting layer at large velocity bins were contaminated by the range sidelobe of the echo below."

- c. Lines 177- 187.

The description of the PDF powers is not described well. I do not understand what analysis techniques are being performed in this paragraph.

Sorry for the unclear description in the original manuscript. We have added the determination of PDF$_{thresh}$ in Appendix B.

- d. Figure 4 is confusing.

To present the results to the reader more clearly, we modified Fig. 4 in the original manuscript (Fig. 5 in the revised manuscript) to show the probability distributions of spectral power of range bins at 2.4 km, 5.01 km, and 6.6 km, which respectively represent the liquid precipitation, Doppler spectra contaminated by range sidelobe, and solid precipitation. The following descriptions have been added to the revised

manuscript:

"An interesting feature of the range sidelobe caused by pulse compression is that its spectral power is much flatter than cloud and precipitation signals. Figure 2a (figure 5a in the manuscript) shows the probability density functions (PDFs) of received spectral power at 2.4 km, 5.01 km, and 6.6 km, which respectively represent the liquid precipitation, Doppler spectrum contaminated by range sidelobe, and solid precipitation. It can be seen that the range bins contaminated by range sidelobe have different spectral power distributions. For Doppler spectra without the sidelobe contamination, the spectral powers corresponding to maximum probability are relatively large. In contrast, for the sidelobe-contaminated Doppler spectrum, its maximum probability corresponds to small power close to the noise level and is mostly below 15 dB above the noise level. A closer look into the radar Doppler spectra at 5.01 km (Fig. 6a) shows that the strong PDF peak is explained by the relatively flat range sidelobe signals. Here, we introduce a parameter spectral power threshold ($S_{thresh}$) to distinguish the range sidelobe from meteorological signals."

- e. While the manuscript describes the excess power above the melting layer, the manuscript does not address the excess power in the range gates below the melting layer. The manuscript should include a range sidelobe correction for range gates below the melting layer.

  The reviewer is correct that the range sidelobe caused by pulse compression technology appears in both the upper and lower range gates of the target bin. The theoretical peak sidelobe ratio (the ratio of the main lobe peak power to the highest sidelobe peak power) is 36 dB and 30 dB for mode 2 and mode 4, respectively. Our statistics (Fig. D1) show that the sidelobe signals are usually below -20 dB. Since the reflectivity enhancement in the melting layer usually does not exceed 10 dB (Li et al., 2020), the sidelobe contamination in rain is not significant. However, the fall velocity of snow is much slower than rain drops. Namely, no meteorological signals present in the range of 3 ~ 10 m/s and the sidelobe signal becomes evident.

- 5. The text on Line 223 states, "This effect leads to the underestimation of V, which is critical in the merging process, and Ze." And line 235 states, "In Fig. 8a1, significant biases of Delta Z and Delta V can be identified, and Delta V increases with Delta Z." There are several issues:
- a. Fig. 8a1 does not show Delta V increasing with Delta Z.
- b. Fig. 8a1 does not show that coherent integration in mode 1 is causing a bias in Z and V relative to mode 3. In fact, the reflectivity difference shown in Fig. 8a1 is of the wrong sense with mode 3 having a smaller reflectivity than mode 1 (Delta Z = mode3 - mode1) Also, there is not a velocity difference between mode1 and mode3 in Fig. 8a1.
  We are sorry for the wrong figure labels in the original manuscript. The figure

labels have been corrected in the revised manuscript.

- c. The manuscript needs to describe the expected differences in Z and V due to coherent integrations and then verify these expectations with the observations.

  In this work, we aim to develop a new framework to generate merged spectral moments. In this section, we want to show that due to the coherent integration Ku-band mode 2 data is still applicable in spectral merging, while Ka-band mode 2 data should be used with caution.

  The statistics of differences in Z and V were to assess whether the coherent integration has a significant impact on Z and V, given V is used in the following spectral merging process. The relation of deltaZ and deltaV depends not only on coherent integration but also on the size distribution of hydrometeors which may be simulated with various parameters. We agree that it would be interesting to look into this in a separate study.

- 6. Figure 12. There is still "clutter" in the moments shown in the left-hand panels. The clutter is present when there is no surface precipitation between 2 and 3 km height and between 20:45 and 21:00 LST. The clutter needs to be removed from this figure or text must be included describing why the clutter is still in this figure. See comment #2, the manuscript needs to verify that the "clutter" in Fig. 12 is due to either non-meteorological scattering or due to cloud particle scattering.

  We thank the reviewer for raising this question. After checking the spectra data, we have found that this is due to the different sensitivities of different observing modes. If the clutter signal is detected only by the most sensitive mode, then the previous method will fail to filter it out because there are no signals from another mode to compare it with.

  In the revised manuscript, we have improved the clutter removal method by adding a step which compares the width of a spectral mode (see Fig. 3 in revised manuscript) if the signal is present in the most sensitive mode only. A new paragraph has been added in Section 3.1 clutter mitigation:

  "It should be noted that this method relies on observations recorded at different observing modes. However, the sensitivities of different modes are not identical. Therefore, if the clutter is presented in the most sensitive mode (e.g., mode 2) only, it cannot be filtered out with the $|\Delta S|$ method. In this case, the width of valid meteorological spectral mode is assumed to be longer than 2 m s$^{-1}$, otherwise it is attributed to clutter. We are aware that Shupe et al (2004) have used a width of 0.448 m s$^{-1}$ to identify supercooled liquid water. We have tried this value, but the width of clutter present in this dual-wavelength radar system easily exceeds 1 m s$^{-1}$ (Fig. 2). Actually, the selection of the spectrum width is similar with the use of a

signal-to-noise ratio (SNR) value in noise-removal. Higher SNR means a stricter noise-removal but higher chance of losing valid signals. We have tested the width of 1, 1.5, 2, and 3 m s$^{-1}$ (visual inspection, not shown), and found that 2 m s$^{-1}$ can effectively remove clutter signals for both radars though very light precipitation (detected by the most sensitive mode only) can be removed as well. Admitting this potential issue, it suffices the application in rainfall. In addition, for clouds with highly variable reflectivity, the presented algorithm may mislabel them as clutter according to our assumption that meteorological signals are coherent in a round of observation (28s)."

**Suggestions**

- 7. Line 43. Define 'non-meteorological clutter'. Is the manuscript referring to reflection from stationary targets (e.g., buildings or poles) or moving targets (e.g., insects, birds, or power lines moving in the wind)?
  We replace the term "non-meteorological clutter" with "non-meteorological signal", which includes signals from stationary targets and moving targets.

  Line 45 has been rewritten as "1) The contamination of non-meteorological signals. The non-meteorological echoes produced by stationary targets (e.g., buildings, trees or terrain) and moving targets (e.g., insects, birds, or power lines moving in the wind) are unwanted but often detected by radar. Narrow-beam-width antenna…"

- 8. Table 1. Include the radar operating mode names to table (i.e., boundary layer mode, cirrus mode, etc)
  The names of operating modes have been added to Table 1.

- 9. Table 1. Include the transmitted pulse length (time and distance) and number of code bits in each mode.
  The pulse lengths of different modes have been added to Table 1.

  In this paper, our Ka/Ku-band cloud radar use the linear frequency modulation (LFM) technique for signal modulation, which is different from phase coding, so we cannot add the number of code bits to the Table. We have added the description of radar signal modulation in the introduction to the radar in Section 2.

  "To improve both the radar detection performance and range resolution, Linear Frequency Modulation was used to widen the signal bandwidth when transmitting pulses in modes 2 and 4 at both Ka- and Ku-band. But, the matched pulse compression filter output exhibits sidelobe behavior, making the power of range sidelobe appear in the wrong range gates." has been added to Section 3.2 in the revised manuscript.

- 10. Figures 12 and 13. Both figures show a gap in cloud structure near 2 km between 20:00 and 20:30 LST. It looks like the mode3 (right panel) shows low

level precipitation that does not extend above 1 km and another mode is detecting precipitation above its blind zone near 2 km. The text should mention the vertical gaps in precipitation features and they are due to blind zone issues and different sensitivities of the modes.

To clarify this, these sentences have been added to the revised manuscript:

"Additionally, due to the differences in sensitivity and the blind zones between different modes, mode 2 with the highest sensitivity only participate in the data merging above 2 km at Ku-band, so it can be seen that there is a gap in cloud structure at 2 km which occurs at Ku-band" (Fig. 15 in the revised manuscript).

---

## Referee Report (RR1)

**Review of amt-2022-169 (Revised submission)**

The authors addressed the issues mentioned in the first review.

In particular, the following points, in my opinion, significantly improve the quality of the manuscript.

- The proposed method undergoes a better evaluation:
  - The spectral width produced by the proposed method is compared with the one measured by a C-band radar.
  - A quantitative evaluation of the impact clutter removal and sidelobe mitigation has been performed.
  - The removal of the range sidelobe artifacts is compared with a different algorithm from the literature (Liu and Zheng, 2019) in section 3.2 and Appendix C.
- The clutter mitigation (section 3.1) is explained in more detail, with a dedicated appendix to describe the choice of the threshold $|\Delta S|$.
- The removal of the range sidelobe artifacts is now illustrated by a clearer figure. The reasons behind the existence of these interference lines are also better explained.
- The application of the mode merging to the Ka-band is shown in an additional figure and briefly discussed in the text, clarifying one of the comments for the previous review.

These large modifications are accompanied by many smaller ones, consisting mostly of clarifications and improvements in the English language of the text.

I recommend the article for publication after addressing the following minor issues.

**Specific issue**

- Section 5.3.1
  As briefly mentioned in the introduction of the review, I think that the quantitative evaluation of the clutter removal is a useful addition that improves the quality of the manuscript.
  In my opinion, however, some of the terms used in the comparison should be renamed to better reflect the true nature of the evaluation presented in this subsection.
  In particular, the name "true data" for the median of the decluttered products (section 5.3.1) may be misleading.
  These measurements labeled as "true data" are not a set of independent observations of better quality (as could be, for example, the variables from another radar with better sensitivity). Instead, they are simply a combination of the best products that the proposed algorithm is able to generate. Therefore, I would suggest to re-phrase some parts of this section, to highlight that the results presented here are a measure of the impact of the decluttering in each mode, but not an estimation of how much closer the processed data are to the true meteorological signal.

**Technical comments**

- Lines 113-116
  The information on cross-calibration is a useful addition. For completeness, I would add the values of the reflectivity offsets computed.

  I also have a small question regarding the computation of the offsets: are these offsets computed on unprocessed data?
  If this is the case, what would the effect of the not-yet removed artifacts (clutter, side-lobes) be on this offset? Would the spectral power of the non-meteorological signal be included in the reflectivity of the unprocessed data, affecting the value of the offset, given the difference in the artifacts between the different modes undergoing the cross-calibration?
  I would expect the effect of the non-meteorological signal to be small, but you could check whether it can be truly ignored by re-computing the offset on your data after processing and then comparing this second offset with the one you computed from the unprocessed ones. I do not think that this additional check should be included in the manuscript, but it could be useful to the authors to verify the cross-calibration.

- Line 165
  The whole explanation of the choice of the threshold on $|\Delta S|$ is a great addition to the manuscript, providing an answer to one of the specific comments mentioned in the previous review.
  Regarding the whole explanation, I have only one small technical comment: the existence of "Appendix A" (detailing the analysis conducted on the $|\Delta S|$ distribution) is only mentioned in parenthesis after referring to the figure A1. Moreover, the appendix is not referred to as "Appendix A" but only as "Appendix".
  I would explicitly mention in the main text that the analysis is provided in the appendix, and I would refer to the latter as "Appendix A", to avoid confusion with the other appendices.

- Lines 259-269
  The addition of a comparison with the algorithm from Liu and Zheng (2019) illustrates the advantage of using the new method proposed by the authors.
  However, I find the last part of Section 3.2 difficult to read, due to the constant references to a figure from the Appendix.
  In my opinion, it would be better to relegate the discussion on the figure to the Appendix, mentioning only a summarized version of it in the text and referring to Appendix C for more details. Alternatively, if the authors want to keep the explanation in the main text, I would move Fig. C1 to the main text as one of the figures of Section 3.2.
  In both cases, I think that "Appendix C" also needs to be mentioned in the main text (similarly to what I wrote in the previous comment regarding Appendix A).

- Line 361
  Why the comparison is done specifically with mode 3? Is it because of its smaller blind range? In my opinion, it could be useful to add a brief explanation behind the choice of this mode in the text.
  I also noticed that other modes (e.g. mode 4, in line 368) are mentioned in the text as a target for the comparison. Why is mode 4 not mentioned alongside mode 3 at the beginning of the section?

- Line 371
  As for Appendix A and C, I would mention Appendix D explicitly in the text.

- Lines 374-375
  "Namely, no meteorological signals present in the range of [...]"
   I believe that the phrase is missing an "are", and it was supposed to be:
   "Namely, no meteorological signals are present in the range of [...]"

- Line 395
  What would be the results if another variable (e.g. reflectivity factor) was used for the comparison? In case you tried the comparison, would it show any improvements linked with the removal of spurious side lobes (which I would expect to be responsible for a slight overestimation of the reflectivity in the unprocessed data), or is the effect too small to be seen?

- Line 418
  Is the median here (and in line 415) performed for each range gate and time step separately? If this is the case, I am confused by the usage of the median as the metric, since it would be the median between only two values (at each gate and time step), and in that case, I think the average would be a better choice.

- Line 425
  The "1dbZ" may be a bit misleading here. For mode 2 the improvement is 0.36, for mode 3 it is 0.8 and for mode 4 it is 0.65. I would re-phrase this statement more accurately as:
  "[...] the SD for the reflectivity at Ku-band is reduced by a value between 0.36 and 0.8 dB after imposing the clutter removal algorithm."
  The difference between two reflectivity factors in dBZ is expressed in dB, so the unit for the difference should be changed.

- Lines 440 – 441
  In my opinion, Appendix B (and D, if not introduced previously) should be introduced explicitly here, with a short phrase detailing the content/objective of each of them.

- Line 467
  Since the applicability of the method in snowfall has not been explicitly shown in the manuscript, I would change the phrase stating clearly that the correct functioning of the algorithm in snowfall is expected but has not been proven yet.

- Line 470 – Appendix A
  I suggest the addition of a short text in Appendix A for explaining the context of Figure A1.

- Line 480
  Is the standard deviation here computed in a similar way as in section 5.3.1?
  In general, I would expand slightly this Appendix, explaining the procedure more extensively, and clearly stating the objective of the comparison (i.e. finding the value of alpha that minimizes the SD).

- Line 499 – Appendix C
  Same comment as for Appendix A.

- Line 513
  How is the height of the peak and sidelobe determined?
  From the figures in the manuscript I expected the sidelobe to span multiple range gates, is its height set as the average of all heights affected? I think that the procedure should be briefly explained in the appendix.

- Line 515
  The terms "main lobe peak power" and "sidelobe peak power" should be explained. Is "sidelobe peak power" the same as the term $S_{peak}$ previously introduced?

---

## Author Response (AR2)

The authors addressed the issues mentioned in the first review.

In particular, the following points, in my opinion, significantly improve the quality of the manuscript.

- The proposed method undergoes a better evaluation:
  ◦ The spectral width produced by the proposed method is compared with the one measured by a C-band radar.

  ◦ A quantitative evaluation of the impact clutter removal and sidelobe mitigation has been performed.

  ◦ The removal of the range sidelobe artifacts is compared with a different algorithm from the literature (Liu and Zheng, 2019) in section 3.2 and Appendix C.
- The clutter mitigation (section 3.1) is explained in more detail, with a dedicated appendix to describe the choice of the threshold $|\Delta S|$.
- The removal of the range sidelobe artifacts is now illustrated by a clearer figure. The reasons behind the existence of these interference lines are also better explained.
- The application of the mode merging to the Ka-band is shown in an additional figure and briefly discussed in the text, clarifying one of the comments for the previous review.

These large modifications are accompanied by many smaller ones, consisting mostly of clarifications and improvements in the English language of the text. I recommend the article for publication after addressing the following minor issues.

The authors would like to thank the reviewer for all the comments on the manuscript, and all of them will be addressed in some manner. Please see the point-to-point response below in blue color.

**Specific issue**

- Section 5.3.1

  As briefly mentioned in the introduction of the review, I think that the quantitative evaluation of the clutter removal is a useful addition that improves the quality of the manuscript.

  In my opinion, however, some of the terms used in the comparison should be renamed to better reflect the true nature of the evaluation presented in this subsection.

  In particular, the name "true data" for the median of the decluttered products (section 5.3.1) may be misleading.

  These measurements labeled as "true data" are not a set of independent observations of better quality (as could be, for example, the variables from another radar with better sensitivity). Instead, they are simply a combination of the best products that the proposed algorithm is able to generate. Therefore, I would suggest to re-phrase some parts of this section, to highlight that the results presented here are a measure of the impact of the decluttering in each mode, but not an estimation of how much

closer the processed data are to the true meteorological signal.

We thank the reviewer for the good suggestion. Given that the standard deviation was calculated relative to a result that we consider to be relatively accurate, the name "true data" has been replaced by "reference data".

**Technical comments**

- Lines 113-116

The information on cross-calibration is a useful addition. For completeness, I would add the values of the reflectivity offsets computed.

I also have a small question regarding the computation of the offsets: are these offsets computed on unprocessed data?

If this is the case, what would the effect of the not-yet removed artifacts (clutter, side-lobes) be on this offset? Would the spectral power of the non-meteorological signal be included in the reflectivity of the unprocessed data, affecting the value of the offset, given the difference in the artifacts between the different modes undergoing the cross-calibration?

I would expect the effect of the non-meteorological signal to be small, but you could check whether it can be truly ignored by re-computing the offset on your data after processing and then comparing this second offset with the one you computed from the unprocessed ones. I do not think that this additional check should be included in the manuscript, but it could be useful to the authors to verify the cross-calibration.

Yes, the reflectivity offsets between different modes were computed on unprocessed data. The weak and stable precipitation cases were selected and observations from mode 2 were used as a reference due to high sensitivity, then the reflectivity difference was computed. For each profile, the median of the reflectivity difference was recorded and the values of offset were determined as the median of all the recorded results.

We did check the offset before and after the decluttering, and the clutter has very minimal impact on the calculation of the offset. This is expected, since the occurrence of non-meteorological signals is much smaller than rain.

The values of the reflectivity offsets have been added to the revised manuscript.

"For both radars, the reflectivity observations at mode 2 were used as the reference to calibrate radar data at other modes. The reflectivity offsets are 3.8 dB (mode 2 - mode 3) and -3.6 dB (mode 2 - mode 4) at Ka-band, respectively. For the Ku-band radar, these values are 7.5 dB (mode 2 – mode 1), -1.0 dB (mode 2 – mode 3) and -2.9 dB (mode 2 – mode 4), respectively."

- Line 165

The whole explanation of the choice of the threshold on |ΔS| is a great addition to the manuscript, providing an answer to one of the specific comments mentioned in the previous review.

Regarding the whole explanation, I have only one small technical comment: the

existence of "Appendix A" (detailing the analysis conducted on the |ΔS| distribution) is only mentioned in parenthesis after referring to the figure A1. Moreover, the appendix is not referred to as "Appendix A" but only as "Appendix".

I would explicitly mention in the main text that the analysis is provided in the appendix, and I would refer to the latter as "Appendix A", to avoid confusion with the other appendices.

*Appendix A has been explicitly mentioned in the revised main text.*

- Lines 259-269

The addition of a comparison with the algorithm from Liu and Zheng (2019) illustrates the advantage of using the new method proposed by the authors.

However, I find the last part of Section 3.2 difficult to read, due to the constant references to a figure from the Appendix.

In my opinion, it would be better to relegate the discussion on the figure to the Appendix, mentioning only a summarized version of it in the text and referring to Appendix C for more details. Alternatively, if the authors want to keep the explanation in the main text, I would move Fig. C1 to the main text as one of the figures of Section 3.2.

In both cases, I think that "Appendix C" also needs to be mentioned in the main text (similarly to what I wrote in the previous comment regarding Appendix A).

*To make the article clear and easy to read, we have moved the analysis in the last part of Section 3.2 to Appendix C, and a summary clarification was added to the text.*

*"Furthermore, we have compared this algorithm with the threshold method (Liu and Zheng, 2019), all the results and analysis are included in Appendix C."*

- Line 361

Why the comparison is done specifically with mode 3? Is it because of its smaller blind range? In my opinion, it could be useful to add a brief explanation behind the choice of this mode in the text.

I also noticed that other modes (e.g. mode 4, in line 368) are mentioned in the text as a target for the comparison. Why is mode 4 not mentioned alongside mode 3 at the beginning of the section?

*We agree with the reviewer. The comparison with mode 4 has been added in the revised manuscript.*

- Line 371

As for Appendix A and C, I would mention Appendix D explicitly in the text.

*The text has been revised to "In addition, we have calculated statistics of the power*

leakage to range sidelobe, and the results for Ku-/Ka-band radars are given in Appendix D (Fig. D1)."

- Lines 374-375

"Namely, no meteorological signals present in the range of [...]"

I believe that the phrase is missing an "are", and it was supposed to be:

"Namely, no meteorological signals are present in the range of [...]"

Corrected.

- Line 395

What would be the results if another variable (e.g. reflectivity factor) was used for the comparison? In case you tried the comparison, would it show any improvements linked with the removal of spurious side lobes (which I would expect to be responsible for a slight overestimation of the reflectivity in the unprocessed data), or is the effect too small to be seen?

We did give thinking on this. However, the radar frequencies are different, and different radars suffered from different wet radar radome and rain attenuation. It is rather challenging to do the attenuation correction for the profiles of Ka and Ku band radar reflectivity observations.

- Line 418

Is the median here (and in line 415) performed for each range gate and time step separately? If this is the case, I am confused by the usage of the median as the metric, since it would be the median between only two values (at each gate and time step), and in that case, I think the average would be a better choice.

We agree with the reviewer. It has amended to average. In addition, we have fixed a mistake in our code and updated the values in Table 3.

- Line 425

The "1dbZ" may be a bit misleading here. For mode 2 the improvement is 0.36, for mode 3 it is 0.8 and for mode 4 it is 0.65. I would re-phrase this statement more accurately as: "[...] the SD for the reflectivity at Ku-band is reduced by a value between 0.36 and 0.8 dB after imposing the clutter removal algorithm."

The difference between two reflectivity factors in dBZ is expressed in dB, so the unit for the difference should be changed.

Thank you very much for this suggestion, the statement has been re-phrased to "the SD for the reflectivity at Ku-band is reduced by a value between 0.36 and 0.8 dB after imposing the clutter removal algorithm."

- Lines 440 – 441

In my opinion, Appendix B (and D, if not introduced previously) should be introduced explicitly here, with a short phrase detailing the content/objective of each of them.

Appendix B was introduced in Line 245 and Appendix D was introduced in Line 387.

The text has been revised as "Radar observations from 4.5 to 6 km are used for the assessment, and the results for the Ku-band radar are given in Appendix B (see Tab. B1 for details). Since the signals associated with sidelobe are relatively weak (Fig. D1 in Appendix D) […]"

- Line 467

Since the applicability of the method in snowfall has not been explicitly shown in the manuscript, I would change the phrase stating clearly that the correct functioning of the algorithm in snowfall is expected but has not been proven yet.

The statement has been revised to "[…], the applicability of the presented framework in snowfall is expected but has not been proven yet."

- Line 470 – Appendix A

I suggest the addition of a short text in Appendix A for explaining the context of Figure A1.

"The meteorological signals with a height of 2 ~ 3 km and Doppler velocity of 2 ~ 5 m s-1 were statistically analyzed to determine the appropriate |ΔS|." has been added in the revised manuscript.

- Line 480

Is the standard deviation here computed in a similar way as in section 5.3.1?

In general, I would expand slightly this Appendix, explaining the procedure more extensively, and clearly stating the objective of the comparison (i.e. finding the value of alpha that minimizes the SD).

Yes, the standard deviation was computed in a similar way as in Section 5.3.1. To make it clear, we added some descriptions in the revised manuscript.

"A similar quantitative evaluation can be made to find the appropriate value of α to maximize the sidelobe mitigation. Since the Doppler spectra observations at mode 3 for both radars are not affected by the sidelobe effect, they are used as the "reference data" at both Ku- and Ka-band. That is,

$$X_{Ku/Ka,ref} = X_{Ku/Ka,M3}^{mitigated} \qquad (B1)$$

Then, the standard deviation between spectral moments with different α and observations at mode 3 was calculated through Eq. (B1) and compared. Radar observations between 4.5 km and 6 km were evaluated, and the results for Ku- and Ka-band radars are given in Tab. B1 and B2, respectively."

- Line 499 – Appendix C

Same comment as for Appendix A.

All the analysis of appendix C has been moved from the main text to the appendix.

- Line 513

How is the height of the peak and sidelobe determined?

From the figures in the manuscript I expected the sidelobe to span multiple range gates, is its height set as the average of all heights affected? I think that the procedure should be briefly explained in the appendix.

The height of peak spectral power refers to the height of the range gate with maximum spectral power for each velocity bin in a spectra profile, and the height of sidelobe refers to the height of the range gate that is identified contaminated by range sidelobe. The procedure has been briefly explained to the appendix.

"This appendix shows how much power is leaking into the range sidelobes. For a given velocity bin in a spectra profile, the maximum spectral power is denoted as $S_{peak}$, and the corresponding height is $H_{peak}$. Then, the spectral power of sidelobe is denoted as $S_{sidelobe}$ and the height as $H_{sidelobe}$. The difference between $S_{peak}$ and $S_{sidelobe}$ and the corresponding $H_{peak}$ and $H_{sidelobe}$ were analyzed."

- Line 515

The terms "main lobe peak power" and "sidelobe peak power" should be explained. Is "sidelobe peak power" the same as the term $S_{peak}$ previously introduced?

The peak sidelobe ratio measures the waveform after pulse compression, it is different from the term $S_{peak}$.

These terms have been clarified in the revised manuscript.

"The theoretical peak sidelobe ratio (the ratio of the main lobe peak power to the highest sidelobe peak power) depends on the transmitted waveform after pulse compression, and is 36 dB and 30 dB for mode 2 and mode 4, respectively."